# UNIVERSALRAG:
# RETRIEVAL-AUGMENTED GENERATION OVER CORPORA OF DIVERSE MODALITIES AND GRANULARITIES

## ABSTRACT

Retrieval-Augmented Generation (RAG) has shown substantial promise in improving factual accuracy by grounding model responses with external knowledge relevant to queries. However, most existing approaches are limited to a text-only corpus, and while recent efforts have extended RAG to other modalities such as images and videos, they typically operate over a single modality-specific corpus. In contrast, real-world queries vary widely in the type of knowledge they require, which a single type of knowledge source cannot address. To this end, we introduce UniversalRAG, designed to retrieve and integrate knowledge from heterogeneous sources with diverse modalities and granularities. Specifically, motivated by the observation that forcing all modalities into a unified representation space derived from a single aggregated corpus causes a modality gap, where the retrieval tends to favor items from the same modality as the query, we propose modality-aware routing that dynamically identifies the most appropriate modality-specific corpus and performs targeted retrieval within it. Also, beyond modality, we organize each modality into multiple granularity levels, enabling fine-tuned retrieval tailored to the complexity and scope of the query. We validate UniversalRAG on 8 benchmarks of multiple modalities, showing superiority over modality-specific and unified baselines.

## 1 INTRODUCTION

Large Language Models (LLMs) have demonstrated remarkable performance across various tasks, and have been widely adopted in services to assist users in everyday life (Anil et al., 2023; OpenAI, 2024). Yet, LLMs often generate factually incorrect or misleading information, especially on topics they were less or not exposed to during training (Zhang et al., 2023; Huang et al., 2025). To address this, Retrieval-Augmented Generation (RAG) has emerged as a promising approach, which allows the model responses to be grounded in the query-relevant knowledge retrieved from external knowledge sources, enhancing factual accuracy (Lewis et al., 2020; Gao et al., 2024; Chen et al., 2024a).

Despite its effectiveness, existing RAG approaches are typically designed for a single corpus and modality, limiting their ability to address queries that require diverse knowledge sources. In practice, as shown in Figure 1, user queries vary widely in the type of knowledge they require: some are best answered using text (e.g., surface-level facts and definitions), others demand visual understanding from images (spatial relations of objects), and yet others require temporal reasoning supported by videos (step-by-step instructions with dynamic scenes). Yet, the field of RAG primarily originates with a textual corpus (Lewis et al., 2020; Jiang et al., 2023; Yan et al., 2024), and although recent efforts have expanded it to modalities beyond text (such as images and videos) (Riedler & Langer, 2024; Abootorabi et al., 2025; Jeong et al., 2025), existing RAG methods individually are typically modality- and corpus-specific; therefore, they may be suboptimal to serve as a universal, one-for-all framework that can flexibly handle the wide range of queries, whose knowledge requirements vary.

In this work, we present UniversalRAG, a novel RAG framework that brings together knowledge distributed across multiple modality-specific corpora, and leverages them to generate grounded responses to queries in a universal workflow. To operationalize this, one straightforward approach might be to aggregate all entries from the collected, heterogeneous knowledge corpora, and embed them into a unified space using a multimodal encoder (which is typically trained to align inputs from

Figure 1: Illustration of different RAG approaches. (A) RAG with Single Modality struggles to handle queries requiring modalities other than one in the corpus; (B) Single Granularity lacks flexibility in granularity, resulting in overly fine or overly coarse information; (C) Single Unified Corpus causes modality gaps that bias retrieval toward the modality of the query; (D) Our UniversalRAG overcomes these limitations via a modality- and granularity-aware routing mechanism over diverse corpora.

different modalities if they are semantically similar). However, despite such alignment efforts, we find that this strategy suffers from modality gaps (Zhang et al., 2025; Bolya et al., 2025; Wang et al., 2024b), the tendency that inputs are clustered based on their modality rather than their semantic meaning (visualized in Figures 2 and 7). As a result, retrieval becomes biased toward knowledge sources that share the same modality as the query, overlooking relevant content from other modalities.

To address this challenge, instead of relying on a unified embedding space that forces all modalities into the shared representation, we take a different direction: introducing a *modality-aware routing strategy*. Specifically, UniversalRAG dynamically determines the most suitable knowledge source to retrieve from, based on the modality requirement of the given query, then routes the retrieval process to the corresponding modality-specific corpus. It is worth noting that this strategy not only sidesteps modality gaps by avoiding every cross-modal comparison but also enables seamless integration of new modalities by extending the routing logic without modifying existing modality-specific retrievers.

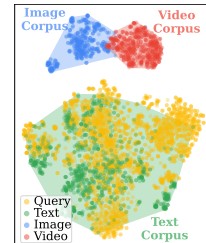

Figure 2: t-SNE result of unified embedding space.

Beyond modality, another important angle is data granularity (the size or unit of each entry in the corpus), which impacts both retrieval precision and generation quality (Chen et al., 2024b; Zhong et al., 2025), since different queries benefit from different levels of granularity even within the same modality: overly fine-grained entries can dilute context, while overly coarse ones may bundle unrelated information. For example, complex analytical questions may require full documents or videos, while simple factoid questions are better served with a single paragraph or short video clip.

To accommodate this aspect, we further break down each modality into multiple granularity levels, organizing them into distinct corpora: textual documents are additionally segmented into paragraphs and stored in a paragraph-level corpus, and similarly, full-length videos are divided into short clips and stored, while images are kept intact since they are inherently piecemeal. Overall, with these modality- and granularity-aware corpora (including paragraphs, documents, images, clips, and videos) in place, as well as an additional no-retrieval option to efficiently handle straightforward queries (that require no external knowledge), our UniversalRAG dynamically routes each query to the most relevant knowledge source, ultimately supporting the diverse information needs of real-world users.

We validate UniversalRAG on 8 benchmark datasets spanning diverse modalities (Yang et al., 2018; Kwiatkowski et al., 2019; Chen et al., 2020; Hendrycks et al., 2021; Chang et al., 2022; Wang et al., 2024a; Jeong et al., 2025). It outperforms all baselines by large margins on average, demonstrating its effectiveness in handling diverse types of queries in realistic scenarios. Moreover, UniversalRAG improves overall efficiency by considering the appropriate levels of granularity (to avoid unnecessary use of lengthy documents or videos), and even maintains robustness on out-of-distribution datasets.

## 2 METHOD

We present UniversalRAG, which retrieves knowledge from multi-modal, multi-granularity corpora.

### 2.1 PRELIMINARIES

**Large Vision Language Models**   Let us first define LLMs, which take an input sequence of tokens $x = [x_1, x_2, \ldots, x_n]$ and generate an output sequence of tokens $y = [y_1, y_2, \ldots, y_m]$, formalized

as follows: $\boldsymbol{y} = \texttt{LLM}(\boldsymbol{x})$, where $x_i$ and $y_i$ are represented in text. Building on top of LLMs, Large Vision-Language Models (LVLMs) extend their capability to support multimodal understanding by incorporating visual encoders (Bai et al., 2023; Chen et al., 2024c; Liu et al., 2024; Li et al., 2024a; Chen et al., 2025; Bai et al., 2025), allowing them to process both the textual and visual inputs. Formally, similar to LLMs, LVLMs can be functionalized as follows: $\boldsymbol{y} = \texttt{LVLM}(\boldsymbol{x})$, whose input token $x_i$ is extended to either textual or visual. However, despite the fact that they are extensively trained, LVLMs themselves are limited to their parametric knowledge, and often struggle with queries that require (fine-grained or up-to-date) information, less or not exposed during training.

**Retrieval-Augmented Generation**   To address the aforementioned limitations of using only the parametric knowledge internalized in models themselves, RAG has been widely used, whose core idea is to retrieve query-relevant information from a large corpus and incorporate it into the generation process. Formally, in response to a query $\boldsymbol{q}$, a retrieval model $\mathcal{T}$ fetches the relevant context $\boldsymbol{c}$ from a corpus $\mathcal{C}$, formalized as follows: $\boldsymbol{c} = \mathcal{T}(\boldsymbol{q}; \mathcal{C})$ where $\boldsymbol{c} \in \mathcal{C}$. Then, in the subsequent generation step, the LVLM generates a response $\boldsymbol{a}$ conditioned on both the query and retrieved context, denoted as follows: $\boldsymbol{a} = \texttt{LVLM}(\boldsymbol{q}, \boldsymbol{c})$. However, most existing RAG approaches are restricted to retrieving from a single corpus consisting of entries from a single modality (such as only the textual documents), limiting their ability to handle diverse queries with knowledge requirements that vary across them.

## 2.2 UNIVERSALRAG

We now turn to introduce UniversalRAG, a novel RAG framework that dynamically identifies and routes queries to the most appropriate modality and granularity of knowledge, for targeted retrieval.

**Challenges in Multi-Corpus Retrieval**   To accommodate the diverse knowledge needs of real-world queries, which may involve heterogeneous sources spanning different modalities, we consider a set of modality-specific corpora, each containing items of a particular type, denoted by $\mathcal{C}_m$ for modality $m$. A straightforward strategy is to aggregate all corpora into a unified corpus $\mathcal{C}_{\texttt{unified}} = \bigcup_{m \in M} \mathcal{C}_m$ and embed all items into a shared space using a multimodal encoder, as for retrieval over a single corpus: $\boldsymbol{c} = \mathcal{T}(\boldsymbol{q}; \mathcal{C}_{\texttt{unified}})$. However, we find this approach suffers from the modality gap (Figures 2 and 7), where queries, being textual, align more closely with elements in the text corpus regardless of the modality required. Therefore, instead of forcing all heterogeneous elements into the unified corpus, we propose selectively engaging the most relevant, modality-specific corpora needed for queries.

**Modality-Aware Retrieval**   To sidestep the modality gap issue (introduced by handling all modalities over the unified space), we instead propose to break down the overall retrieval process into two subsequent stages: 1) identifying the most relevant set of modalities for the given query; and 2) performing targeted retrieval within the selected modality-specific corpora. Specifically, instead of merging all modality-specific corpora into a single corpus, we preserve each corpus in its original form with an independent embedding space. After that, to direct queries to their best-aligned knowledge sources (based on their modality-specific needs), we introduce a routing module $\mathcal{R}$ that dynamically predicts the subset of modalities best suited for a query $\boldsymbol{q}$, yielding $M_{\boldsymbol{q}} = \mathcal{R}(\boldsymbol{q}) \subseteq M$. Retrieval is then restricted to the corresponding corpora $\{\mathcal{C}_m \mid m \in M_{\boldsymbol{q}}\}$, using any off-the-shelf retriever $\mathcal{T}_m$ tailored to each modality, thereby avoiding the modality gap issue present in a unified space. Proposition 1 formally states that modality-aware retrieval achieves higher effectiveness than unified embedding when modality bias is present. We provide the proof for all propositions in Appendix C.

**Proposition 1.** *Let the similarity score in the unified embedding space of $\mathcal{C}_{\text{unified}}$ be defined as*

$$s(\boldsymbol{q}, \boldsymbol{c}) = \alpha \cdot \mathbf{1}\{m(\boldsymbol{q}) = m(\boldsymbol{c})\} + \beta \cdot r(\boldsymbol{q}, \boldsymbol{c}) + \varepsilon,$$

*where $\alpha > 0$ is a modality bias, $m(\cdot)$ denotes the modality, and $r(\cdot)$ measures semantic relevance. If $\alpha$ is sufficiently large relative to the variance of $r$, the probability of retrieving items from the required modality $m^*(\boldsymbol{q})$ is less than under modality-aware routing followed by within-modality retrieval.*

However, while this routing principle mitigates the modality gap, organizing corpora solely by modality might still be suboptimal, as different queries require varying levels of granularity.

**Granularity-Aware Retrieval**   To accommodate the varying complexity and information scope of different queries, we extend UniversalRAG to operate not only across modalities but also across different levels of granularity within each modality. To be specific, rather than treating each modality-specific corpus as a flat collection of items, we organize it into representations at multiple resolutions,

enabling retrieval to target either fine-grained details or broader context as required by the query. For example, a video corpus may be accessed at the level of short clips for focused questions or as full-length videos when comprehensive understanding is required. Building on this richer organization of corpora, the routing module $\mathcal{R}$ expands its prediction space to include modality-granularity pairs best suited to a query, as well as a no-retrieval option for cases where external context is unnecessary:

$$\mathcal{R} : Q \to \{\varnothing\} \cup \mathcal{P}\left( \bigcup_{m \in M} \{m\} \times G_m \right),$$

where $Q$ is the space of queries, $M$ is the set of modalities, $G_m$ is the set of granularities available for modality $m$, and $\mathcal{P}(\cdot)$ denotes the power set. Once the router predicts the relevant pairs, retrieval is performed over the corresponding corpora, using retrievers specialized for each modality to obtain the relevant content $c$. Finally, the LVLM generates the answer $a$ with $c$, customized to the modality and granularity for the query, thereby enabling the universal, one-for-all RAG framework. Proposition 2 states that adapting granularity to the query always yields strictly higher expected response quality.

**Proposition 2.** *Let $F(Q; m, g)$ be the expected response quality when retrieving from modality $m$ using granularity $g$. If there exist queries $q_1, q_2$ and granularities $g_f, g_c$ such that*

$$F(q_1; m, g_f) > F(q_1; m, g_c) \quad and \quad F(q_2; m, g_c) > F(q_2; m, g_f),$$

*then the routing policy that assigns $g_f$ for $q_1$ and $g_c$ for $q_2$ achieves strictly higher expected quality than any fixed-granularity choice.*

### 2.3 ROUTER IMPLEMENTATION STRATEGIES IN UNIVERSALRAG

A key component of UniversalRAG is the router, which is responsible for determining the optimal modality and granularity of knowledge for the given query. We consider two operational strategies.

**Training-based Router** To tailor the available model for the routing task, we first consider training it to predict the appropriate modality–granularity pair for each query. However, since ground-truth labels (for the modality and granularity the query should be routed to) are not available, we leverage inductive biases in existing benchmarks, mapping each dataset to routing targets that match its task characteristics (e.g., clip retrieval for localized events vs. full-video retrieval for long-range video understanding), allowing us to automatically curate a labeled corpus without manual annotation. We then train a lightweight model, such as DistilBERT (Sanh et al., 2019), to serve as the router. At inference time, to account for cross-modality needs, the router may output multiple configurations when their confidence scores exceed a threshold, enabling cross-modality or multi-granularity retrieval when necessary, while standard single-modality queries remain routed to their single best match.

**Training-free Router** Alternatively, we also explore a training-free approach that leverages the broad knowledge and robust reasoning capabilities of modern frontier models, such as Gemini (Anil et al., 2023). Instead of learning from labeled data, the model is directly prompted to act as routers. Specifically, we first design the prompt template (used to elicit routing), which describes the objective and includes examples demonstrating how different types of queries correspond to specific retrieval targets (See Figure 8 for details). Then, at inference, the model is prompted with this template to predict the most suitable modality-granularity pairs from a predefined set. This eliminates the need for supervised labels or task-specific training, offering the flexibility to adapt to new tasks and domains.

## 3 EXPERIMENT

### 3.1 EXPERIMENTAL SETUP

We explain the experimental setup, including datasets, models, metrics, and implementation details.

**Datasets** To evaluate UniversalRAG, we compile a comprehensive benchmark with various datasets for RAG, spanning seven different modalities and granularities. Specifically, for the no-retrieval, we use **MMLU** (Hendrycks et al., 2021), which assesses the capability of models themselves without requiring external sources. For the text RAG setting, we incorporate representative datasets such as **Natural Questions (NQ)** (Kwiatkowski et al., 2019), designed for single-hop RAG with paragraphs as the retrieval units; and **HotpotQA** (Yang et al., 2018), which targets multi-hop RAG with documents

(or sets of paragraphs) as the retrieval units. For the table RAG, we include **HybridQA** (Chen et al., 2020), a benchmark that requires reasoning over tables combined with additional text sources. For the image RAG, we consider **WebQA** (Chang et al., 2022), whose subset consists of queries that require grounding in external images. Lastly, for the video RAG, we use three datasets: **LVBench** (Wang et al., 2024a), whose queries target short or localized segments of video content; and **VideoRAG-Wiki** (Jeong et al., 2025) and **VideoRAG-Synth** (Jeong et al., 2025) that often consist of queries requiring comprehension of long-form (or complete) videos. Please see Appendix A for more details.

**Knowledge Corpora** To support diverse RAG scenarios with different modalities and granularities, we consider their corresponding corpora. Specifically, for the text RAG, in addition to the Wikipedia paragraph corpus compiled from Karpukhin et al. (2020), we also use the corpus of multi-paragraph documents following Jiang et al. (2024b) to build it by aggregating Wikipedia paragraphs. For the table corpus, we collect tables from the HybridQA benchmark. For the image, we use the corpus from the WebQA, consisting of images. Lastly, for the video, we construct two corpora (according to granularity): initially designing the video corpus by collecting full-length videos from LVBench and VideoRAG datasets, and segmenting them into multiple clips to construct the clip-level corpus. Together, these corpora define the seven routing pathways: **None**, **Paragraph**, **Document**, **Table**, **Image**, **Clip**, and **Video**. We provide additional details on corpus construction in Appendix A.

**Methods** We compare UniversalRAG to a diverse set of 12 baselines, grouped into four categories. The first is **Naïve**, which directly answers queries without retrieving external knowledge. In addition, the group of **Unimodal RAGs** includes **ParagraphRAG**, **DocumentRAG**, **TableRAG**, **ImageRAG**, **ClipRAG**, and **VideoRAG** methods, which retrieve information only from their respective modality-specific corpora and leverage it for response generation. The third group, **Unified Embedding Multimodal RAGs**, includes approaches that utilize the single embedding space for RAG, such as **UniRAG** (Sharifymoghaddam et al., 2025) and **GME** (Zhang et al., 2025) that perform retrieval over multimodal data (such as text and images) by representing them into the shared space; **InternVideo2**[1] (Wang et al., 2024b) and **PE$_{core}$** (Bolya et al., 2025) that use multimodal encoders (trained to align different modalities) for representing videos as well as text and images. Lastly, **All** is included in the last group of **Multi-corpus Multimodal RAGs**, which performs retrieval over all the modality-specific corpora and incorporates the retrieved results into the LVLM for response generation. Notably, as UniversalRAG is operationalized with different routing strategies, we implement its several variants: training-based variants, **UniversalRAG (DistilBERT)** and **UniversalRAG (T5-Large)**, which train DistilBERT (Sanh et al., 2019) and T5-Large (Raffel et al., 2020) with the automatically constructed routing dataset to return the single-target prediction, and a training-free variant, **UniversalRAG (GPT-4.1)**, prompts GPT-4.1 (OpenAI, 2024) to select the most prominent modality-granularity pair. A further variant, **UniversalRAG (Cross-GPT-4.1)**, also leverages GPT-4.1 but is prompted to allow the selection of multiple modality-granularity pairs, enabling retrieval from diverse sources for queries that benefit from evidence across modalities. Finally, we include an oracle setup (**Oracle**), which routes each query to its ideal corpus, non-comparable with others.

**Evaluation Metrics** We report results with standard metrics. For datasets with multiple-choice questions, we report **Top-1 Accuracy (Acc)**, the proportion of questions answered correctly. For short-answer datasets, we use **Exact Match (EM)** and **F1**, which respectively measure exact agreement and word-level overlap between predictions and references. For datasets with longer free-form answers, we use **ROUGE-L**, which captures the longest common subsequences between the prediction and reference (Lin, 2004), and **BERTScore**, which assesses their semantic similarity (Zhang et al., 2020). We report the average score by averaging first within each modality, then across modalities.

**Implementation Details** For generations, we employ multiple LVLMs, including InternVL2.5-8B (Chen et al., 2025), Qwen2.5-VL-7B-Instruct (Bai et al., 2025), and Phi-3.5-Vision-Instruct (Abdin et al., 2024). Also, to take advantage of UniversalRAG in routing the retrieval process to the modality-specific corpus, we use modality-specific encoders: bge-large-en-v1.5 (Xiao et al., 2024) for text, InternVideo2 (Wang et al., 2024b) for vision, and dense row-level embedding (Ji et al., 2025) with the text encoder for tables, retrieving the nearest entries via cosine similarity over their embedding space. Lastly, for the router, we train it (for training-based variants) for 5 epochs with a learning rate of 2e-5 for DistilBERT and for 10 epochs with a learning rate of 3e-5 for T5-Large, selected

---

[1] InternVideo2 also serves as the visual encoder for UniversalRAG. Unless otherwise specified, the term "InternVideo2" refers to the unified embedding baseline.

Table 1: Results of diverse RAG methods with InternVL2.5-8B by modalities. Our UniversalRAG is represented by the colored cells. **Bold** indicates the best performance and underline indicates the second-best, among UniversalRAG approaches. R-L and BERT refer to ROUGE-L and BERTScore.

| Models | MMLU | NQ | | HotpotQA | | HybridQA | | WebQA | | LVBench | VideoRAG-Wiki | | VideoRAG-Synth | | Avg |
|---|---|---|---|---|---|---|---|---|---|---|---|---|---|---|---|
| | Acc | EM | F1 | EM | F1 | EM | F1 | R-L | BERT | Acc | R-L | BERT | R-L | BERT | |
| Naïve | 64.50 | 24.71 | 38.11 | 12.92 | 20.87 | 0.86 | 4.91 | 40.63 | 90.30 | 28.60 | 15.74 | 84.20 | 14.93 | 85.73 | 31.16 |
| ParagraphRAG | 64.50 | 35.14 | 47.89 | 14.45 | 23.05 | 7.43 | 10.98 | 37.25 | 89.77 | 28.80 | 13.92 | 83.68 | 22.18 | 87.29 | 33.28 |
| DocumentRAG | 51.50 | 23.57 | 32.66 | 19.71 | 28.49 | 6.71 | 10.67 | 28.92 | 87.45 | 28.80 | 13.28 | 83.75 | 18.51 | 86.12 | 28.80 |
| TableRAG | 54.50 | 9.43 | 15.42 | 9.19 | 14.49 | 7.29 | 11.17 | 31.33 | 88.68 | 27.03 | 12.11 | 83.21 | 18.77 | 86.31 | 24.80 |
| ImageRAG | 54.50 | 23.57 | 32.96 | 13.11 | 20.18 | 1.29 | 5.53 | 46.50 | 91.32 | 31.64 | 17.26 | 83.79 | 20.72 | 87.02 | 30.68 |
| ClipRAG | 53.50 | 13.86 | 21.82 | 9.38 | 16.51 | 1.29 | 4.95 | 39.53 | 90.27 | 35.36 | 18.76 | 86.38 | 27.37 | 89.34 | 28.41 |
| VideoRAG | 59.50 | 14.43 | 22.98 | 9.95 | 16.95 | 1.29 | 5.03 | 40.08 | 90.51 | 33.59 | 19.23 | 86.35 | 28.23 | 89.45 | 29.36 |
| UniRAG (Sharifymoghaddam et al., 2025) | 57.50 | 16.14 | 27.49 | 9.57 | 16.49 | 0.43 | 3.61 | 43.98 | 90.89 | 25.27 | 15.86 | 83.95 | 24.75 | 88.22 | 28.14 |
| GME (Zhang et al., 2025) | 60.00 | 15.57 | 26.65 | 10.53 | 17.95 | 4.71 | 9.63 | 45.16 | 90.04 | 26.15 | 17.28 | 84.89 | 26.33 | 88.50 | 29.96 |
| InternVideo2 (Wang et al., 2024b) | 58.00 | 17.43 | 27.79 | 10.33 | 17.76 | 1.00 | 3.20 | 45.12 | 91.09 | 27.82 | 15.66 | 83.78 | 24.43 | 88.13 | 29.01 |
| PE_core (Bolya et al., 2025) | 60.50 | 16.57 | 27.34 | 9.76 | 16.67 | 1.29 | 4.19 | 44.19 | 90.84 | 28.31 | 15.91 | 83.98 | 23.63 | 87.99 | 29.20 |
| All | 58.50 | 28.86 | 41.72 | 16.17 | 26.63 | 5.57 | 10.13 | 40.39 | 90.32 | 32.62 | 15.33 | 85.03 | 26.87 | 88.92 | 33.60 |
| **UniversalRAG (DistilBERT)** | 62.50 | 34.86 | 47.08 | 18.56 | 26.96 | 7.86 | 12.04 | 46.32 | 91.28 | **35.65** | 19.23 | 86.35 | 28.23 | 89.45 | 36.82 |
| **UniversalRAG (T5-Large)** | 63.00 | 35.43 | 47.71 | **18.95** | **27.56** | 7.14 | 12.00 | 46.43 | 91.27 | 34.38 | 19.23 | 86.35 | 28.20 | **89.49** | **36.95** |
| **UniversalRAG (GPT-4.1)** | **65.00** | 34.86 | 47.60 | 15.89 | 23.84 | 8.57 | 12.13 | 44.74 | 91.00 | 31.15 | 13.95 | 83.68 | 22.49 | 84.71 | 35.27 |
| **UniversalRAG (Cross-GPT-4.1)** | 63.50 | 35.86 | 47.86 | 15.98 | 24.21 | **10.71** | **15.57** | **48.13** | **95.02** | 30.36 | 16.27 | 84.30 | 25.39 | 88.92 | 36.25 |
| Oracle | 64.50 | 35.14 | 47.89 | 19.71 | 28.49 | 12.00 | 17.16 | 46.50 | 91.32 | 35.65 | 18.79 | 86.38 | 27.45 | 89.35 | 38.31 |

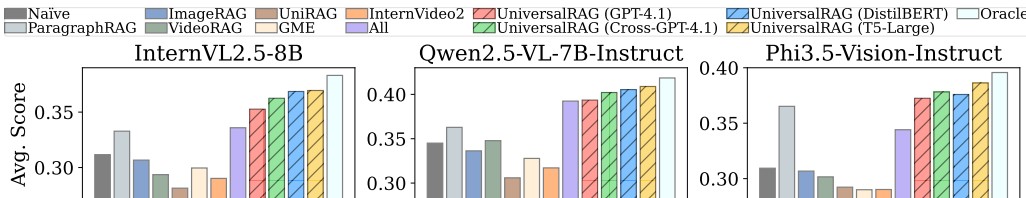

Figure 3: Comparison of averaged evaluation results across different RAG methods and LVLMs.

based on validation accuracy; meanwhile, for the training-free variant, we prompt GPT-4.1 with task objectives and examples, as shown in Figure 8. Further details are provided in Appendix B.

## 3.2 EXPERIMENTAL RESULTS AND ANALYSES

Now we present the overall results across diverse RAG scenarios spanning multiple modalities and levels of granularity, followed by a detailed analysis of the observed performance improvements.

**Overall Results** We present the modality- and granularity-specific results in Table 1, along with the averaged results with different LVLMs in Figure 3, from which we observe that UniversalRAG consistently achieves the best performance on average. Specifically, in Table 1, the results compared against the unimodal RAG baselines corroborate our hypothesis that retrieving from the modality (or granularity) that aligns best with the information needs of queries achieves the highest accuracy; however, mismatches between the query and retrieval source results in significant degradation, which supports our claim that considering the diverse modalities in the universal workflow is necessary for realistic RAG. Also, the level of granularity within each modality affects performance, suggesting that fine-grained retrieval and generation are necessary. In addition to them, UniversalRAG significantly outperforms another category of unified embedding multimodal RAG baselines (forcing all modalities into a single space), confirming the issue of the modality gap inherent within them (Figures 2 and 7). Lastly, when compared with the 'All' baseline (within the multi-corpus multimodal RAG category), which results in suboptimal performance due to the inclusion of noise from irrelevant modalities in generation, our UniversalRAG remains effective. Its strong performance is due to its core idea around modality-specific routing, enabling the selective retrieval from the most relevant modality and granularity for each query, yielding performance gains despite using several corpora.

**Effectiveness of Cross-Modal Retrieval** While many queries can be addressed by using a single, most prominent modality, certain tasks benefit from integrating evidence across multiple modalities. For instance, WebQA involves visually grounded questions that pair text with images, while HybridQA requires reasoning that spans both structured tables and accompanying textual sources. In such cases, UniversalRAG (Cross-GPT-4.1), which enables retrieval from multiple modality-granularity sources, demonstrates clear advantages over unimodal variants by aggregating cross-modal evidence. As shown in Table 1, the cross-modal variant achieves overall improvements across benchmarks, with large gains on WebQA and HybridQA. These results highlight the value of cross-modal retrieval in scenarios when single-modality evidence is insufficient, but also the flexibility of UniversalRAG to support both single- and cross-modal retrieval; however, they also suggest that current benchmarks underrepresent such cross-modal queries, suggesting the need for a richer evaluation suite.

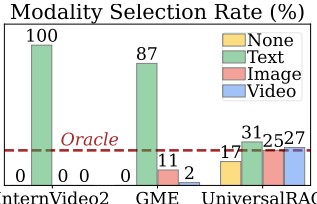

Figure 4: Distribution of the retrieved data modalities.

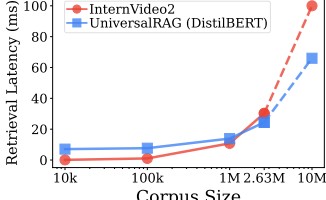

Figure 5: Retrieval latency per query across corpus sizes.

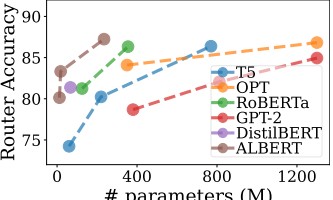

Figure 6: Router accuracy with varying the router model size.

Table 2: Modality accuracy (in corpus selection) and recall of retrieved items for retrieval methods.

| Models | Modality Acc | Recall R@1 | R@3 | R@5 |
|---|---|---|---|---|
| UniRAG | 25.00 | 0.02 | 0.05 | 0.06 |
| GME | 41.29 | 23.01 | 34.29 | 40.80 |
| InternVideo2 | 25.00 | 1.87 | 2.44 | 4.01 |
| PE$_{core}$ | 25.00 | 0.98 | 1.34 | 1.72 |
| **UniversalRAG** (DistilBERT) | 83.64 | 29.73 | 45.19 | 53.24 |
| **UniversalRAG** (T5-Large) | **87.71** | **32.01** | **46.68** | **54.09** |
| **UniversalRAG** (GPT-4.1) | 68.85 | 19.92 | 32.77 | 37.80 |

Table 3: Performance across different numbers of granularity (#Gn) for two router models.

| Models | #Gn | HotpotQA EM | F1 | LVBench Acc |
|---|---|---|---|---|
| GPT-4.1 | 1 | 11.00 | 21.91 | 29.29 |
| | 2 | **15.89** | 23.84 | 31.15 |
| | 3 | 15.79 | **24.11** | 31.15 |
| | 4 | 15.60 | 23.64 | **31.83** |
| Gemini 2.5 Flash | 1 | 14.45 | 22.99 | 31.48 |
| | 2 | 17.61 | 25.79 | 32.57 |
| | 3 | 17.27 | 24.95 | 32.81 |
| | 4 | **17.70** | **25.86** | **33.60** |

**Effectiveness of Modality Routing** To investigate the benefit of our routing method, we compare the distribution of retrieved modalities among InternVideo2, GME, and UniversalRAG (DistilBERT), summarized in Figure 4. Using 200 sampled queries per benchmark and normalizing distributions for balance, we find that InternVideo2 retrieves only text (including tables), while GME exhibits a similar bias toward text regardless of the actual modality required for the given query. This highlights how the modality gap in the unified embedding space makes retrieval ineffective. However, UniversalRAG distributes retrieval more evenly across modalities, demonstrating that the query router effectively mitigates modality bias and adaptively directs queries to the most suitable knowledge source. This also results in high modality retrieval accuracy – the accuracy with which the correct modality (i.e., none, text, image, or video) is retrieved – which directly translates to high recall of the retrieved items, as shown in Table 2. Specifically, while GME achieves comparable recall on text and image corpora, its inability to accurately retrieve from the correct modality leads to lower recall on multimodal corpora that include videos. Yet, UniversalRAG, with trained routers, consistently retrieves from the correct modality, enabling it to achieve higher retrieval recall than baselines across all scenarios.

**Effectiveness of Multigranularity** Given the observed benefits of corpus selection in Table 1, we further investigate its impact beyond modality choice by comparing UniversalRAG at varying levels of granularity. Table 3 shows that incorporating granularity-aware corpus selection leads to consistent performance gains by avoiding the retrieval of context that is either insufficient (e.g., a short paragraph lacking key entities for multi-hop reasoning) or excessive (e.g., a full video when only a short clip is relevant), both of which can hinder accurate response generation. Also, as additional granularity levels are introduced, we observe further improvements in some cases, though gains are not strictly monotonic across tasks, reflecting the trade-off between context sufficiency and noise. In the meantime, we adopt a binary level of granularity in our main experiments to strike a balance between effectiveness and efficiency. Results with trained router variants are reported in Table 11.

**Retrieval Efficiency of Modality-Specific Retrieval** Beyond accuracy, UniversalRAG also improves efficiency by reducing the search space: it leverages modality- and granularity-aware routing to restrict retrieval to only the most relevant sources, instead of querying a unified embedding index that aggregates all modalities into a single mega-corpus. Also, the overhead for routing is small as this cost is outweighted at scale by the reduced search space, leading to sub-linear latency growth as corpus size increases, as shown in Figure 5. Specifically, UniversalRAG eventually achieves lower latency than unified embedding methods at large corpus sizes, with the gap widening further at very large scales (e.g., beyond 10M entries). This scalability makes UniversalRAG a practical solution for real-world applications, where corpora are significantly larger than our experimental settings.

**Analysis on Router Size** To examine whether the routing cost (while already small) can be further reduced by using smaller models as routers without sacrificing accuracy, we train six models (Liu et al., 2019; Radford et al., 2019; Sanh et al., 2019; Lan et al., 2020; Raffel et al., 2020; Zhang et al.,

Table 4: Results of UniversalRAG and baselines on out-of-domain dataset with InternVL2.5-8B.

| Models | TruthfulQA Acc | TriviaQA EM | TriviaQA F1 | LaRA R-L | LaRA BERT | Visual-RAG R-L | Visual-RAG BERT | Cinepile Acc | Avg |
|---|---|---|---|---|---|---|---|---|---|
| Naïve | 64.68 | 49.47 | 57.92 | 23.15 | 87.62 | 6.24 | 80.98 | 30.76 | 33.88 |
| ParagraphRAG | 58.73 | 54.61 | 65.14 | 20.23 | 86.48 | 4.74 | 80.77 | 30.07 | 33.88 |
| DocumentRAG | 28.73 | 39.94 | 44.73 | 25.18 | 86.83 | 4.34 | 81.14 | 32.64 | 26.68 |
| ImageRAG | 57.85 | 45.23 | 52.50 | 21.40 | 87.09 | 7.31 | 82.32 | 34.03 | 33.35 |
| ClipRAG | 51.01 | 31.62 | 42.40 | 19.64 | 87.50 | 6.92 | 81.32 | 35.63 | 29.59 |
| VideoRAG | 47.34 | 33.59 | 43.82 | 19.89 | 87.19 | 7.04 | 81.42 | 37.43 | 29.47 |
| UniRAG (Sharifymoghaddam et al., 2025) | 55.70 | 39.64 | 47.88 | 19.66 | 86.47 | 5.20 | 80.67 | 31.60 | 29.16 |
| GME (Zhang et al., 2025) | 54.94 | 54.31 | 65.12 | 19.28 | 86.31 | 5.64 | 81.14 | 30.14 | 33.82 |
| InternVideo2 (Wang et al., 2024b) | 52.15 | 35.70 | 45.01 | 21.28 | 86.83 | 4.31 | 80.47 | 30.76 | 28.92 |
| PE$_{core}$ (Bolya et al., 2025) | 55.82 | 39.94 | 48.67 | 18.78 | 86.01 | 4.80 | 80.67 | 30.97 | 28.54 |
| All | 45.82 | 28.74 | 41.63 | 19.14 | 87.01 | 6.02 | 80.77 | 36.60 | 29.07 |
| **UniversalRAG** (DistilBERT) | **56.08** | 42.06 | 51.74 | 21.03 | 86.98 | **7.35** | 82.30 | **35.63** | 32.65 |
| **UniversalRAG** (T5-Large) | 55.57 | 43.72 | 52.04 | 21.38 | **87.02** | 7.31 | **82.32** | **35.63** | 32.63 |
| **UniversalRAG** (GPT-4.1) | 54.54 | **53.25** | **62.33** | **23.77** | 86.88 | 7.12 | 82.29 | 35.14 | **36.17** |
| Oracle | 64.68 | 55.52 | 64.85 | 25.18 | 86.83 | 7.31 | 82.32 | 37.71 | 38.26 |

Table 5: Input token efficiency with respect to RAG performance on text and video datasets.

| Models | Avg # Tokens ↓ | Avg Score ↑ |
|---|---|---|
| *Text-based Generation Scenarios* | | |
| ParagraphRAG | **182** | 35.47 |
| DocumentRAG | 3912 | 30.57 |
| **UniversalRAG** (DistilBERT) | 2126 | **37.02** |
| *Video-based Generation Scenarios* | | |
| ClipRAG | **2154** | 24.37 |
| VideoRAG | 8466 | 25.07 |
| **UniversalRAG** (DistilBERT) | 6236 | **26.48** |

Table 6: Router accuracy and generation performance across retrieval methods on two settings.

| Models | In-Domain Router Acc | In-Domain Avg Score | Out-Domain Router Acc | Out-Domain Avg Score |
|---|---|---|---|---|
| Random | 14.29 | 28.91 | 16.67 | 29.99 |
| **UniversalRAG** (DistilBERT) | 81.38 | 36.86 | 40.10 | 32.65 |
| **UniversalRAG** (T5-Large) | **86.38** | **36.95** | 49.63 | 32.63 |
| **UniversalRAG** (GPT-4.1) | 51.20 | 35.27 | **63.86** | **36.17** |
| Ensemble (Confidence-based) | 79.32 | 36.71 | 62.44 | 35.98 |
| Ensemble (Majority Voting) | 85.29 | 36.90 | 51.32 | 34.88 |

2022) ranging from 12M to 1.3B parameters and measure router accuracy. As Figure 6 shows, router accuracy increases with larger model sizes within each architecture, suggesting the scalability of our routing approach. However, even compact models such as ALBERT achieve strong performance with only 12M parameters, indicating that compact models can be effectively utilized in UniversalRAG.

**Generation Efficiency of Multigranularity** We hypothesize that the multigranular retrieval of UniversalRAG is also superior in generation efficiency compared to baselines, for which we present the average length of the input tokens (including retrieved data and query) with average scores for text and video datasets in Table 5 (where we sample 32 frames for full videos and 8 frames for clips). Fine-granularity baselines process shorter information during inference but underperform compared to UniversalRAG, as the retrieved information is often insufficient to accurately answer the query; meanwhile, coarse-granularity baselines provide more context but at the cost of substantially longer inputs. UniversalRAG achieves the best of both worlds: it consistently outperforms coarse-granularity baselines with fewer tokens, and it surpasses fine-granularity baselines by retrieving just enough context to answer the query. For example, in text datasets, UniversalRAG achieves a 6.5% higher average score than the Document baseline, while using only about half as many input tokens. With granularity-aware retrieval, UniversalRAG can balance performance and computation efficiency.

**Generalizability on Out-of-Domain Datasets** While results on in-domain datasets demonstrate the strong performance of UniversalRAG relative to baseline methods, particularly when using trained routers, we also evaluate its ability to generalize to out-of-domain (OOD) tasks. Table 4 presents the full comparison between UniversalRAG and the baselines on OOD benchmarks. The results show that trained routers underperform on TriviaQA and LaRA, where they either fail to select the correct granularity or encounter task types that were largely unseen during training. In contrast, the training-free router exhibits robust performance even in these challenging scenarios. For the remaining benchmarks, a pattern similar to the in-domain setting emerges: UniversalRAG outperforms unified embedding baselines, with trained routers outperforming training-free routers.

Table 6 summarizes UniversalRAG's average performance across in-domain and OOD settings. Interestingly, the trend reverses in the OOD scenario: the training-free router provides more robust performance, while the trained router experiences a more noticeable decline. These results underscore

Table 7: Case study comparing unimodal RAGs with fixed granularity to UniversalRAG (Ours).

| Question | Who finishes first in the Men's 100M Round 1 Heat 5 during the London 2012 Olympics, featuring Usain Bolt and Yohan Blake? (A) Su BingTian (B) Usain Bolt (C) Asafa Powell (D) Tyson Gay   **Answer : (C)** | | |
|---|---|---|---|
| **TextRAG** | **Retrieved:** former 100 m world champion, Zhanna Pintusevich-Block) of Total Sports Management. On July 28, 2006, he announced a deal with Nike that will run through to the 2012 Summer Olympics in London. On July 11, 2006, at the Grand Prix in Lausanne ... | **ImageRAG** | **Retrieved:** |
| | **Response:** (B) ✗ | | **Response:** (B) ✗ |
| **VideoRAG** | **Retrieved:** (Timestamp Range: 00:00~38:26) | **Ours** | **Routed to:** Clip  **Retrieved:** (Timestamp Range: 25:57~29:22) |
| | **Response:** (B) ✗ | | **Response:** (C) ✓ |

the advantage of leveraging the training-free router's inherent model knowledge, which enables stronger generalization to unseen tasks and makes it particularly effective in OOD conditions.

**Ensemble Strategy for Robust Routing**   Building on our findings of the trade-off between the high in-domain accuracy of trained routers and the strong out-of-domain generalization of the training-free router, we further explore a novel ensemble strategy to leverage their complementary strengths. In particular, we propose two ensemble strategies: confidence-based ensemble and majority voting. In a confidence-based ensemble, the prediction of the trained router (DistilBERT) is used if its confidence score exceeds a predefined threshold; otherwise, the system falls back to the training-free router (GPT-4.1). For majority voting, we adopt the majority answer from three routers (including training-based and free) as a final prediction; if no majority exists, one is selected at random. Table 6 shows that UniversalRAG with the ensemble routers offers a robust middle ground between them, suggesting that it could be beneficial in real-world scenarios with unseen or shifting distributions.

**Case Study**   We present a case study of UniversalRAG in Table 7. The query asks for the winner of Heat 5 in the Men's 100M Round 1 at the London 2012 Olympics. TextRAG and ImageRAG retrieve a paragraph and an image related to the Olympics, but neither provides relevant evidence to answer the question, resulting in incorrect responses. Meanwhile, VideoRAG retrieves the full video of Men's 100M Round 1 at the 2012 Olympics, but struggles to identify the winner of Heat 5 due to the inclusion of irrelevant segments from other heats. However, UniversalRAG selects 'Clip' corpus and retrieves the video clip for Heat 5, enabling the generation model to focus on the specific race mentioned in the query and generate the correct answer. More case studies are shown in Appendix F.

## 4   RELATED WORK

**Large Vision Language Models**   Building on the impressive performance of LLMs (Anil et al., 2023; OpenAI, 2024), recent studies have extended their capabilities to handle visual information. Specifically, Liu et al. (2023) introduces one of the first Large Vision Language Models (LVLMs) by incorporating a CLIP-based (Radford et al., 2021) image encoder, enabling the language model to interpret visual inputs within a shared textual feature space. Subsequently, a variety of LVLMs have been proposed, each integrating different image encoders (Bai et al., 2023; Chen et al., 2024c; Liu et al., 2024), and this line of work has more recently been extended to video data (Li et al., 2024a; Chen et al., 2025; Bai et al., 2025). However, despite improved performance on multimodal benchmarks (Mathew et al., 2021; Yue et al., 2024; Li et al., 2024b; Fu et al., 2024), enabled by larger training datasets and better model architectures, LVLMs still often suffer from hallucinations (Huang et al., 2025), only when they rely solely on parametric knowledge acquired during pretraining.

**Retrieval-Augmented Generation**   To address the aforementioned limitation of parametric-only models, RAG has emerged, incorporating external knowledge during response generation. While conventional RAG methods primarily operate over textual corpora (Lewis et al., 2020; Ram et al., 2023), recent studies have begun to explore RAG over multimodal sources (such as images and videos) (Chen et al., 2022; Riedler & Langer, 2024; Jeong et al., 2025). However, these approaches assume a fixed single-modality retrieval, making them less adaptable to real-world queries that may require information from different modalities. One promising approach is to leverage multimodal encoders (Radford et al., 2021; Wang et al., 2024b; Zhang et al., 2025; Bolya et al., 2025) that can

encode text, images, and videos into a shared embedding space, and Sharifymoghaddam et al. (2025) proposes to retrieve from such a unified embedding space; however, it often struggles to retrieve visual content when queries are text. While other approaches (Cui et al., 2024; Liu et al., 2025a) attempt to retrieve knowledge from all modalities, followed by extra selection mechanisms during or after generation, they incur notable computational overhead. Lastly, adaptive retrieval strategies (Jeong et al., 2024; Islam et al., 2024; Ding et al., 2024; Yao et al., 2024; Tang et al., 2025), proposed to handle diverse query needs, are limited to a single corpus (Zhang et al., 2024; Li et al., 2024c).

**Retrieval Granularity**    While most existing RAG systems operate at a fixed granularity (e.g., full documents, passages, or sentences), real-world queries often require information at varying levels of specificity depending on the knowledge needed, which in turn impacts performance and efficiency in both textual (Chen et al., 2024b; Liu et al., 2025b; Zhong et al., 2025) and video-based retrieval systems (Chen et al., 2023). In contrast, UniversalRAG performs query-level routing across modality and granularity dimensions, enabling retrieval from the most relevant source at the appropriate level.

## 5 CONCLUSION

In this paper, we propose UniversalRAG, a novel RAG framework designed to retrieve from corpora with diverse modalities and granularities. Through a modality- and granularity-aware routing mechanism, UniversalRAG dynamically selects the most suitable knowledge source for each query, effectively addressing the limitations posed by modality gaps and fixed-granularity retrieval. Extensive evaluations across 8 benchmarks demonstrate that UniversalRAG consistently outperforms both modality-specific and unified baselines, showcasing robust performance across diverse modalities. Also, our analyses highlight the importance of fine-grained retrieval and the complementary strengths of training-free and trained routers. These findings demonstrate the potential of UniversalRAG as an adaptive solution for grounding LVLMs with heterogeneous external knowledge, paving the way for a one-for-all RAG solution that unifies the fragmented landscape of existing corpus-specific RAGs.

## ETHICS STATEMENT

The proposed UniversalRAG can be seamlessly integrated with any LVLMs and compatible retrieval corpora, reducing hallucination with the corpus-specific routing. However, there can be potential private, harmful, or biased content present in the retrieved or generated outputs, depending on the nature of the underlying corpora or the internalized knowledge within LVLMs. To mitigate such risks, it is recommended to apply safeguard mechanisms and filtering techniques in retrieval and generation, to ensure the safe and responsible deployment.

## REPRODUCIBILITY STATEMENT

We take several steps to ensure the reproducibility of our work. All experimental details are described in Section 3.1 and Appendix B. The preprocessing pipeline for all datasets, along with benchmark sampling and corpus formulation, is described in Appendix A. Our routing and generation components mainly utilize open-source LLMs and LVLMs, which are fully reproducible, with the exception of training-free routers that are based on closed-source APIs. Lastly, we attach the complete source code in the supplementary materials, covering all stages from data preprocessing to end-to-end evaluation.

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

# A   ADDITIONAL DETAILS ON DATASET

Table 8 provides an overview of all datasets and their corresponding knowledge corpora used in our experiments, including the target modality type as well as the size of the queries and corpora. We divide each dataset into a 3:7 ratio for training and testing. We offer the detail of each dataset below.

## A.1   IN-DOMAIN DATASET

**MMLU**   As a dataset comprising queries that can be answered without the need for retrieval, we use MMLU (Hendrycks et al., 2021), a benchmark that spans a wide range of tasks, including problem-solving abilities (e.g., elementary mathematics, computer science) and world knowledge (e.g., law, world religions). Specifically, we use questions from all tasks in the development split.

**Natural Questions (NQ)**   We also use Natural Questions (Kwiatkowski et al., 2019), a question answering dataset consisting of real user queries issued to the Google search engine, with answers annotated based on supporting Wikipedia articles. We randomly sample 1,000 QA pairs from the dev split, and formulate the text corpus in the same setting as SQuAD, segmenting the Wikipedia corpus into paragraphs of at most 100 words.

**HotpotQA**   HotpotQA (Yang et al., 2018) is a Wikipedia-based QA benchmark, but it contains complex queries that are annotated to reason over multiple articles. We utilize 1,492 randomly sampled QA pairs of the test split. As it requires multi-hop reasoning over multiple documents, we formulate the text corpus by grouping multiple related documents following LongRAG (Jiang et al., 2024b), which can be longer than 4K tokens.

**HybridQA**   HybridQA (Chen et al., 2020) is a benchmark that requires reasoning over both tabular and textual information. Each question is grounded in a Wikipedia table, but often requires linking to associated text information to locate the correct answer. We randomly sample 2,000 QA pairs from the dev split. Unlike the original benchmark, which directly connects tables and textual evidence, we separate them into distinct table and text corpora to better validate our modality-specific routing-based retrieval framework.

**WebQA**   WebQA (Chang et al., 2022) is a benchmark designed to evaluate the ability of LVLMs to reason over multiple sources of information, including both text and images, in an open-domain setting. As the dataset is originally constructed with question-specific retrieval sources that combine text and images, we extract a subset of questions that require only a single image for retrieval. We then further filter these using GPT-4o with the prompt shown in Figure 9 to make sure questions are not grounded to a certain image, resulting in a final set of 2,000 QA pairs.

**LVBench**   LVBench (Wang et al., 2024a) is a benchmark developed for long video understanding, featuring questions generated by annotators based on YouTube videos with an average duration of over one hour. Since the benchmark was originally designed for non-RAG tasks, we rephrase the original text-video interleaved queries into a text-only format to align with our experimental setup using GPT-4o, with video metadata and a prompt (Figure 10). Each query is associated with a specific video and a corresponding time range. Notably, the majority of queries are annotated with timestamps spanning less than five minutes, thereby focusing on short segments within the longer videos. For training, we use these short-timestamp queries as a clip-level dataset.

**VideoRAG**   We also utilize VideoRAG-Wiki and VideoRAG-Synth benchmarks, introduced in VideoRAG (Jeong et al., 2025), which are designed to evaluate RAG over a video corpus. These benchmarks are built on the HowTo100M (Miech et al., 2019) corpus (a large-scale collection of instructional YouTube videos) with queries sourced from WikiHowQA (Bolotova-Baranova et al., 2023) and synthetically generated QA pairs based on the videos. Since they lack timestamp annotations, we employ GPT-4o to identify video-level queries that are better answered through full video retrieval rather than short segments from the ground-truth video, which are then used as a video-level dataset for training the router.

Table 8: Dataset summary. Average corpus length is the mean token count for text corpora and the mean duration for video corpora.

| Dataset | Gold Retrieval | # Queries | Corpus Size | Avg Corpus Length |
|---|---|---|---|---|
| *In-Domain Datasets* | | | | |
| MMLU | None | 285 | - | - |
| Natural Questions | Paragraph | 1,000 | 850k | 100 tokens |
| HotpotQA | Document | 1,492 | 509k | 693 tokens |
| HybridQA | Table | 1,000 | 15k | - |
| WebQA | Image | 2,000 | 20k | - |
| LVBench | Clip/Video | 1,376 | 94 | 3,941s |
| VideoRAG-Wiki | Clip/Video | 374 | 9k | 378s |
| VideoRAG-Synth | Clip/Video | 374 | | |
| *Out-of-Domain Datasets* | | | | |
| TruthfulQA | None | 790 | - | - |
| TriviaQA | Paragraph | 661 | 661k | 100 tokens |
| LaRA | Document | 112 | 34 | 28k tokens |
| Visual-RAG | Image | 374 | 2k | - |
| CinePile | Clip/Video | 1,440 | 144 | 158s |

## A.2 OUT-OF-DOMAIN DATASET

Unlike the in-domain datasets, the out-of-domain datasets are used solely for evaluation to assess the generalizability of our routing approach and consist only of test splits.

**TruthfulQA** TruthfulQA (Lin et al., 2022) includes general knowledge questions designed to test whether LLMs can avoid common false beliefs or misconceptions, on diverse categories, including health, law, and politics. We use the multiple-choice version of the dataset, which includes only a single correct answer per question.

**TriviaQA** TriviaQA (Joshi et al., 2017) is a reading comprehension dataset consisting of trivia questions paired with evidence texts sourced from Wikipedia and the web. To distinguish between queries that require text retrieval and those that do not, we categorize each query based on whether GPT-4o can produce an exact-match answer without access to external text. We randomly sample QA pairs from the dev split. Following the preprocessing strategies used in SQuAD and NQ, all supporting evidence documents are segmented into paragraphs of no more than 100 words.

**LaRA** We also utilize LaRA (Li et al., 2025), which is designed for understanding long-context documents such as academic papers and novels. For our use case, we focus on a subset of these documents, specifically excluding queries on the 'comparison' task, as our goal is RAG, not reading comprehension. Additionally, we slightly reformat the remaining queries to align with a general QA format. Given the length of the source material, each document is treated as a single entry in the document-level corpus.

**Visual-RAG** Visual-RAG (Wu et al., 2025) is a question-answering benchmark designed for visual knowledge-intensive questions, specifically tailored for text-to-image retrieval tasks. We utilize the full set of provided queries but sample five images per category to construct the image retrieval pool, ensuring efficient text-to-image retrieval.

**CinePile** CinePile (Rawal et al., 2024) is a long-video question-answering benchmark that features questions based on movie clips from YouTube. Since the benchmark was originally designed for video understanding tasks rather than RAG, we reformulate each query using the same procedure as LVBench. For each of the 144 available videos, we randomly select 10 questions from the test split. Since CinePile does not provide granularity annotations, we classify the questions into two categories (such as clip-level and full-video-level granularity) using GPT-4o, following the same approach used in VideoRAG.

## B  ADDITIONAL IMPLEMENTATION DETAILS

To effectively leverage both visual and textual information for visual element retrieval, we employ an ensemble approach that combines visual and textual similarity scores with a weighting ratio of 0.8 for visual information. The textual information consists of image captions for images and scripts for videos. To handle long videos, we utilize PySceneDetect (Castellano, 2014), an open-source tool that detects scene boundaries by analyzing content changes (e.g., color histogram differences or threshold-based detection), to segment long videos into shorter clips with an average length of no more than 3 minutes. Moreover, for both the retrieval and generation stages, we uniformly sample 32 frames per video. For baseline models that do not natively support video input, specifically UniRAG (which utilizes CLIP) and GME, we average the embeddings of these sampled frames to obtain a single representative embedding vector. Our experiments are conducted on NVIDIA RTX A6000 GPUs equipped with 48GB VRAM.

## C  PROOF OF PROPOSITIONS

### C.1  PROOF OF PROPOSITION 1

**Proposition 1.** *Let the similarity score in the unified embedding space of $\mathcal{C}_{\text{unified}}$ be defined as*

$$s(\boldsymbol{q}, \boldsymbol{c}) = \alpha \cdot \mathbf{1}\{m(\boldsymbol{q}) = m(\boldsymbol{c})\} + \beta \cdot r(\boldsymbol{q}, \boldsymbol{c}) + \varepsilon,$$

*where $\alpha > 0$ is a modality bias, $m(\cdot)$ denotes the modality, and $r(\cdot)$ measures semantic relevance. If $\alpha$ is sufficiently large relative to the variance of $r$, the probability of retrieving items from the required modality $m^*(\boldsymbol{q})$ is less than under modality-aware routing followed by within-modality retrieval.*

*Proof.* Without loss of generality, we consider the top-1 retrieval case, as the extension to the top-$k$ case follows directly. Let the unified retrieval corpus $\mathcal{C}_{\text{unified}}$ be decomposed into three disjoint sets:

$$S = \{\boldsymbol{c} : m(\boldsymbol{c}) = m(\boldsymbol{q})\}, \quad R = \{\boldsymbol{c} : m(\boldsymbol{c}) = m^*(\boldsymbol{q})\}, \quad O = \mathcal{C}_{\text{unified}} \setminus (S \cup R).$$

Let us assume the scenario where $m^*(\boldsymbol{q}) \neq m(\boldsymbol{q})$ and $S, R \neq \emptyset$. Define

$$X_c := \beta \cdot r(\boldsymbol{q}, \boldsymbol{c}) + \varepsilon_{\boldsymbol{c}},$$

and suppose $\{X_{\boldsymbol{c}}\}_{\boldsymbol{c} \in \mathcal{C}_{\text{unified}}}$ are independent, mean-zero, sub-Gaussian with variance proxy $\sigma^2 = \beta^2 \cdot \text{Var}[r(\boldsymbol{q}, \boldsymbol{c})] + \text{Var}[\varepsilon_{\boldsymbol{c}}]$. Then the similarity scores can be written as

$$s(\boldsymbol{q}, \boldsymbol{c}) = \begin{cases} \alpha + X_{\boldsymbol{c}}, & \boldsymbol{c} \in S \\ X_{\boldsymbol{c}}, & \boldsymbol{c} \in R \cup O. \end{cases}$$

Let

$$M_S = \max_{\boldsymbol{s} \in S} X_{\boldsymbol{s}}, \ M_R = \max_{\boldsymbol{r} \in R} X_{\boldsymbol{r}}, \ M_O = \max_{\boldsymbol{o} \in O} X_{\boldsymbol{o}}.$$

Under unified retrieval, the top-1 item lies in $R$ if and only if

$$M_R \geq \alpha + \max\{M_S, M_O\}.$$

Hence, we can obtain the upper bound of the probability where top-1 retrieval comes from $R$:

$$\mathbb{P}(\mathcal{T}_{\text{unified}}(\boldsymbol{q}; \mathcal{C}_{\text{unified}}) \in R) = \mathbb{P}(M_R \geq \alpha + \max\{M_S, M_O\}) \leq \mathbb{P}(M_R - M_S \geq \alpha). \quad (1)$$

As $\{M_R - M_S \geq \alpha\} \subseteq \cup_{(\boldsymbol{r}, \boldsymbol{s}) \in R \times S}\{X_{\boldsymbol{r}} - X_{\boldsymbol{s}} \geq \alpha\}$, by the union bound we have

$$\mathbb{P}(M_R - M_S \geq \alpha) \leq \sum_{(\boldsymbol{r}, \boldsymbol{s}) \in R \times S} \mathbb{P}(X_{\boldsymbol{r}} - X_{\boldsymbol{s}} \geq \alpha).$$

As $X_{\boldsymbol{r}} - X_{\boldsymbol{s}}$ is sub-Gaussian with variance proxy $2\sigma^2$, the Chernoff bound of the tail probability combined with Equation 1 leads to:

$$\mathbb{P}(\mathcal{T}_{\text{unified}}(\boldsymbol{q}; \mathcal{C}_{\text{unified}}) \in R) \leq |R||S| \exp\left(-\frac{\alpha^2}{4\sigma^2}\right). \quad (2)$$

By contrast, if the retrieval is done at the modality-specific corpus after modality-aware routing with accuracy $r$, the probability where the top-1 item is in $R$ is $r$. Combining this with Equation 2,

$$\mathbb{P}(\mathcal{T}_{\texttt{unified}}(\boldsymbol{q};\mathcal{C}_{\texttt{unified}}) \in R) \leq |R||S| \exp\left(-\frac{\alpha^2}{4\sigma^2}\right) < r = \mathbb{P}(\mathcal{T}_{\mathcal{R}(\boldsymbol{q})}(\boldsymbol{q};\mathcal{C}_{\mathcal{R}(\boldsymbol{q})}) \in R)$$

whenever $\alpha > 2\sigma\sqrt{\frac{\log(|R||S|)}{r}}$. Meanwhile, the right-hand side of Equation 2 decays to 0 as $\alpha/\sigma \to \infty$. Hence, for $\alpha$ large enough relative to the variance of $r$, unified embedding retrieval is strictly worse than retrieving from modality-specific corpus after modality-aware routing. □

*Remark.* Suppose we have very large corpora with size $|R| = |S| = 10^{12}$. In this setting, if $p = 0.8$ and $\sigma = 0.01$, then $\alpha > 2\sigma\sqrt{\frac{\log(|R||S|)}{p}} \simeq 0.17$ is a sufficient condition to make routing-based retrieval more effective than unified embedding retrieval. Since most multimodal encoders exhibit inherent modality bias, this underscores the necessity of modality-aware routing.

### C.2 PROOF OF PROPOSITION 2

**Proposition 2.** *Let $F(Q; m, g)$ be the expected response quality when retrieving from modality $m$ using granularity $g$. If there exist queries $\boldsymbol{q}_1, \boldsymbol{q}_2$ and granularities $g_f, g_c$ such that*

$$F(\boldsymbol{q}_1; m, g_f) > F(\boldsymbol{q}_1; m, g_c) \quad and \quad F(\boldsymbol{q}_2; m, g_c) > F(\boldsymbol{q}_2; m, g_f),$$

*then the routing policy that assigns $g_f$ for $\boldsymbol{q}_1$ and $g_c$ for $\boldsymbol{q}_2$ achieves strictly higher expected quality than any fixed-granularity choice.*

*Proof.* Consider any fixed policy that always uses a single granularity $g \in \{g_f, g_c\}$.

- If $g = g_f$:
$$F(\boldsymbol{q}_1; m, g_f) + F(\boldsymbol{q}_2; m, g_f) < F(\boldsymbol{q}_1; m, g_f) + F(\boldsymbol{q}_2; m, g_c).$$

- If $g = g_c$:
$$F(\boldsymbol{q}_1; m, g_c) + F(\boldsymbol{q}_2; m, g_c) < F(\boldsymbol{q}_1; m, g_f) + F(\boldsymbol{q}_2; m, g_c).$$

In both cases, the sum of response quality with the routing policy that uses $g_f$ for $\boldsymbol{q}_1$ and $g_c$ for $\boldsymbol{q}_2$ is strictly larger than under any fixed granularity $g$. □

## D ADDITIONAL EXPERIMENTAL RESULTS

### D.1 ADDITIONAL RESULTS USING DIFFERENT LVLMS

Table 9 shows detailed generation results of baselines and UniversalRAG models on 8 benchmarks using Qwen2.5-VL-7B and Phi-3.5-Vision-Instruct as generation models. In both settings, UniversalRAG outperforms all baselines and achieves average scores comparable to Oracle. These results demonstrate that UniversalRAG is robust and generalizable in various LVLM generators.

### D.2 ROUTING RESULTS PER DATASET

We present routing results of three routers for each dataset in Table 10. On in-domain datasets, GPT-4.1 often struggles to distinguish between Paragraph and Document RAG queries, and misroutes VideoRAG queries to the textual corpus. Meanwhile, two trained routers show strong classification performance across all in-domain datasets. In out-of-domain datasets, GPT-4.1 generalizes well for most datasets, except for image-based RAG queries. In contrast, trained routers fail to classify the appropriate granularity needed for each query. This is mainly due to the limited diversity of training data, which causes overfitting to seen examples.

Table 9: Results of diverse RAG variants using different LVLMs, including UniversalRAG and baselines, on modality-specific benchmarks.

| | Models | MMLU Acc | NQ EM | NQ F1 | HotpotQA EM | HotpotQA F1 | HybridQA EM | HybridQA F1 | WebQA R-L | WebQA BERT | LVBench Acc | VideoRAG-Wiki R-L | VideoRAG-Wiki BERT | VideoRAG-Synth R-L | VideoRAG-Synth BERT | Avg |
|---|---|---|---|---|---|---|---|---|---|---|---|---|---|---|---|---|
| **Qwen2.5-VL-7B** | Naïve | 73.00 | 17.29 | 25.71 | 18.47 | 25.47 | 3.14 | 7.72 | 61.26 | 94.39 | 29.38 | 14.26 | 83.04 | 10.52 | 84.34 | 34.50 |
| | ParagraphRAG | 72.00 | 39.57 | 50.33 | 17.80 | 24.71 | 8.43 | 12.02 | 49.00 | 92.06 | 27.52 | 14.82 | 83.24 | 11.30 | 84.97 | 36.29 |
| | DocumentRAG | 66.50 | 23.14 | 31.02 | 20.96 | 28.78 | 7.43 | 11.30 | 54.37 | 92.71 | 27.23 | 14.78 | 83.33 | 11.39 | 84.50 | 33.94 |
| | TableRAG | 66.00 | 9.29 | 13.87 | 11.39 | 15.79 | 7.00 | 11.07 | 41.80 | 90.86 | 24.49 | 15.48 | 83.29 | 9.98 | 83.19 | 27.15 |
| | ImageRAG | 68.50 | 16.14 | 23.14 | 16.94 | 23.01 | 1.86 | 5.22 | 64.39 | 94.73 | 30.17 | 16.17 | 83.62 | 13.35 | 85.10 | 33.63 |
| | ClipRAG | 68.50 | 15.14 | 22.69 | 16.46 | 22.86 | 2.71 | 5.59 | 62.78 | 94.38 | 33.50 | 18.39 | 85.04 | 20.53 | 87.75 | 34.43 |
| | VideoRAG | 70.00 | 14.00 | 21.42 | 17.42 | 23.74 | 2.43 | 5.63 | 63.89 | 94.54 | 32.81 | 19.34 | 85.64 | 23.31 | 88.52 | 34.78 |
| | UniRAG (Sharifymoghaddam et al., 2025) | 69.50 | 11.86 | 19.51 | 14.45 | 21.26 | 1.86 | 5.26 | 51.37 | 92.37 | 28.01 | 15.05 | 82.80 | 12.77 | 85.03 | 30.60 |
| | GME (Zhang et al., 2025) | 70.00 | 12.43 | 20.02 | 14.55 | 21.08 | 5.29 | 9.24 | 59.61 | 93.93 | 28.01 | 16.53 | 83.72 | 18.01 | 86.04 | 32.78 |
| | InternVideo2 (Wang et al., 2024b) | 71.50 | 12.29 | 19.81 | 14.35 | 21.11 | 2.00 | 4.42 | 55.64 | 93.07 | 30.14 | 14.97 | 82.83 | 11.38 | 84.16 | 31.71 |
| | PE$_{core}$ (Bolya et al., 2025) | 70.50 | 12.29 | 20.00 | 14.45 | 20.84 | 2.29 | 5.42 | 51.09 | 92.30 | 27.62 | 14.77 | 82.75 | 11.23 | 84.72 | 30.57 |
| | All | 71.00 | 39.00 | 49.86 | 19.04 | 27.56 | 7.29 | 10.59 | 63.89 | 94.48 | 30.85 | 15.64 | 83.62 | 14.23 | 86.03 | 39.24 |
| | **UniversalRAG (DistilBERT)** | 71.50 | 39.00 | 49.45 | 19.62 | 27.41 | 7.86 | 11.84 | 64.12 | 94.70 | 33.20 | 19.34 | 85.64 | 23.45 | 88.53 | 40.54 |
| | **UniversalRAG (T5-Large)** | 72.50 | 39.43 | 49.86 | 20.19 | 28.61 | 8.00 | 12.14 | 64.08 | 94.61 | 33.00 | 19.34 | 85.64 | 23.10 | 88.60 | 40.89 |
| | **UniversalRAG (GPT-4.1)** | 73.50 | 38.00 | 48.39 | 18.37 | 25.17 | 8.57 | 12.56 | 62.01 | 94.32 | 30.95 | 14.82 | 83.25 | 21.88 | 87.80 | 39.36 |
| | **UniversalRAG (Cross-GPT-4.1)** | 72.50 | 38.29 | 48.00 | 18.18 | 25.07 | 10.29 | 14.57 | 66.01 | 96.49 | 32.13 | 15.23 | 83.08 | 21.05 | 86.87 | 40.21 |
| | Oracle | 73.00 | 39.57 | 50.33 | 20.96 | 28.78 | 13.57 | 18.33 | 64.39 | 94.73 | 33.20 | 18.43 | 85.05 | 20.70 | 87.80 | 41.85 |
| **Phi-3.5-Vision-Instruct** | Naïve | 61.00 | 10.43 | 18.49 | 14.26 | 21.01 | 2.29 | 5.57 | 54.01 | 93.01 | 29.58 | 15.94 | 83.64 | 34.58 | 90.66 | 30.95 |
| | ParagraphRAG | 59.00 | 35.57 | 46.59 | 16.36 | 23.56 | 6.86 | 10.29 | 59.18 | 93.51 | 29.87 | 16.91 | 84.84 | 32.28 | 89.84 | 36.53 |
| | DocumentRAG | 52.50 | 16.43 | 24.80 | 17.80 | 25.86 | 6.57 | 10.14 | 57.46 | 93.18 | 29.09 | 14.05 | 84.18 | 33.27 | 90.18 | 32.25 |
| | TableRAG | 47.00 | 5.29 | 9.41 | 11.00 | 15.36 | 5.14 | 8.65 | 55.68 | 92.79 | 28.60 | 15.16 | 83.65 | 30.47 | 89.48 | 26.91 |
| | ImageRAG | 55.50 | 9.86 | 15.73 | 13.68 | 18.70 | 1.29 | 3.13 | 63.25 | 94.13 | 31.15 | 15.16 | 85.02 | 34.18 | 90.32 | 30.69 |
| | ClipRAG | 54.00 | 11.43 | 16.48 | 13.40 | 18.73 | 1.14 | 2.79 | 60.22 | 93.60 | 32.13 | 19.50 | 86.04 | 36.34 | 90.97 | 31.15 |
| | VideoRAG | 53.00 | 9.29 | 15.09 | 13.11 | 17.91 | 1.57 | 3.25 | 59.90 | 93.50 | 32.13 | 19.33 | 86.14 | 36.71 | 90.95 | 30.16 |
| | UniRAG (Sharifymoghaddam et al., 2025) | 54.50 | 5.57 | 13.32 | 11.48 | 18.36 | 1.00 | 4.12 | 58.94 | 92.27 | 28.21 | 16.69 | 84.03 | 35.52 | 90.82 | 29.24 |
| | GME (Zhang et al., 2025) | 54.00 | 5.29 | 13.02 | 11.29 | 17.72 | 3.14 | 6.71 | 56.22 | 93.68 | 27.72 | 18.12 | 84.90 | 36.00 | 90.07 | 29.00 |
| | InternVideo2 (Wang et al., 2024b) | 55.00 | 5.86 | 13.48 | 11.87 | 18.46 | 0.71 | 3.25 | 58.01 | 93.42 | 27.74 | 18.09 | 84.76 | 35.78 | 90.82 | 29.02 |
| | PE$_{core}$ (Bolya et al., 2025) | 54.50 | 5.43 | 13.11 | 11.77 | 18.61 | 1.14 | 4.47 | 56.58 | 93.22 | 26.80 | 16.72 | 83.98 | 35.75 | 90.85 | 29.05 |
| | All | 55.50 | 34.86 | 47.08 | 12.44 | 13.68 | 6.43 | 10.11 | 55.28 | 93.29 | 31.14 | 18.28 | 85.92 | 35.12 | 89.92 | 34.42 |
| | **UniversalRAG (DistilBERT)** | 54.50 | 34.71 | 45.58 | 16.46 | 24.54 | 6.43 | 10.23 | 63.23 | 94.11 | 34.48 | 19.33 | 86.14 | 36.49 | 90.92 | 37.59 |
| | **UniversalRAG (T5-Large)** | 59.50 | 34.00 | 44.86 | 17.22 | 25.74 | 6.86 | 11.00 | 63.26 | 94.02 | 33.00 | 19.33 | 86.14 | 36.69 | 90.86 | 38.63 |
| | **UniversalRAG (GPT-4.1)** | 59.00 | 33.57 | 44.69 | 15.50 | 22.83 | 7.00 | 10.94 | 62.32 | 94.00 | 33.20 | 16.89 | 84.83 | 32.24 | 89.88 | 37.25 |
| | **UniversalRAG (Cross-GPT-4.1)** | 59.00 | 33.43 | 44.29 | 15.91 | 22.49 | 9.71 | 12.14 | 64.11 | 94.25 | 33.99 | 17.25 | 85.01 | 34.07 | 90.13 | 37.82 |
| | Oracle | 61.00 | 35.57 | 46.59 | 17.80 | 25.86 | 11.29 | 15.36 | 63.25 | 94.13 | 34.57 | 19.53 | 86.04 | 36.20 | 90.97 | 39.57 |

Table 10: Routing results across in-domain and out-of-domain datasets. VRAG-Wiki, VRAG-Synth, and Vis-RAG refer to VideoRAG-Wiki, VideoRAG-Synth, and Visual-RAG, respectively.

| Models | Routes | In-Domain Dataset | | | | | | | | Out-of-Domain Dataset | | | | |
|---|---|---|---|---|---|---|---|---|---|---|---|---|---|---|
| | | MMLU | NQ | HotpotQA | HybridQA | WebQA | LVBench | VRAG-Wiki | VRAG-Synth | TruthfulQA | TriviaQA | LaRA | Vis-RAG | CinePile |
| | Routes | 200 | 700 | 1045 | 700 | 1392 | 829 | 374 | 374 | 790 | 661 | 112 | 374 | 1440 |
| **DistilBERT** | None | **84** | 1 | 0 | 0 | 0 | 0 | 0 | 0 | 2 | 1 | 14 | 0 | 0 |
| | Paragraph | 54 | **663** | 132 | 71 | 21 | 1 | 0 | 5 | **642** | 274 | 24 | 0 | 0 |
| | Document | 7 | 10 | **808** | 183 | 8 | 1 | 0 | 0 | 44 | **332** | 5 | 2 | 0 |
| | Table | 11 | 2 | 77 | **429** | 1 | 2 | 0 | 0 | - | - | - | - | - |
| | Image | 3 | 19 | 19 | 6 | **1352** | 7 | 0 | 0 | 16 | 34 | 4 | **371** | 1 |
| | Clip | 7 | 1 | 5 | 9 | 17 | **818** | 0 | 2 | 4 | 8 | 27 | 0 | **1439** |
| | Video | 34 | 4 | 4 | 2 | 1 | 0 | **374** | 367 | 82 | 12 | **38** | 1 | 0 |
| **T5-Large** | None | **149** | 16 | 0 | 0 | 0 | 0 | 0 | 0 | 16 | 5 | 12 | 0 | 0 |
| | Paragraph | 35 | **649** | 39 | 39 | 8 | 2 | 0 | 0 | **638** | **385** | 41 | 1 | 0 |
| | Document | 12 | 24 | **947** | 293 | 10 | 0 | 0 | 0 | 71 | 247 | **43** | 0 | 0 |
| | Table | 0 | 10 | 57 | **359** | 5 | 0 | 0 | 0 | - | - | - | - | - |
| | Image | 0 | 0 | 1 | 5 | **1360** | 5 | 0 | 0 | 8 | 15 | 2 | **373** | 0 |
| | Clip | 0 | 0 | 0 | 0 | 12 | **820** | 0 | 4 | 5 | 3 | 2 | 0 | **1439** |
| | Video | 4 | 1 | 1 | 4 | 5 | 2 | **374** | **370** | 52 | 6 | 12 | 0 | 1 |
| **GPT-4.1** | None | 126 | 58 | 60 | 3 | 60 | 0 | 5 | 19 | **482** | 218 | 1 | 0 | 0 |
| | Paragraph | 46 | **612** | **510** | 200 | 213 | 53 | **368** | **341** | 277 | **427** | 47 | 31 | 6 |
| | Document | 4 | 3 | 357 | **335** | 9 | 27 | 0 | 6 | 30 | 14 | **64** | 0 | 8 |
| | Table | 24 | 21 | 118 | 160 | 23 | 26 | 0 | 0 | - | - | - | - | - |
| | Image | 0 | 3 | 0 | 2 | **1091** | 98 | 1 | 6 | 1 | 2 | 0 | **343** | 25 |
| | Clip | 0 | 3 | 0 | 0 | 4 | **603** | 0 | 0 | 0 | 0 | 0 | 0 | **1362** |
| | Video | 0 | 0 | 0 | 0 | 0 | 22 | 0 | 2 | 0 | 0 | 0 | 0 | 39 |

## D.3 ADDITIONAL RESULTS ON MULTIGRANULARITY

While Table 3 demonstrated a correlation between the number of granularity levels and end-to-end performance using two training-free models, leveraging the flexibility of the approach in scenarios without labeled data. Table 11 extends this by including two training-based variants, comparing the performance with and without granularity. The results consistently show an advantage when granularity is leveraged, showcasing its efficacy across both training-based and training-free approaches.

Table 11: Effect of granularity on the performance. Gn denotes Granularity.

| | | HotpotQA | | LVBench |
|---|---|---|---|---|
| **Models** | **Gn** | EM | F1 | Acc |
| DistilBERT | ✗ | 13.88 | 22.30 | 32.57 |
| | ✓ | **18.56** | **26.96** | **35.65** |
| T5-Large | ✗ | 14.16 | 22.01 | 33.90 |
| | ✓ | **18.95** | **27.56** | **34.38** |
| GPT-4.1 | ✗ | 11.00 | 21.91 | 29.29 |
| | ✓ | **15.89** | **23.84** | **31.15** |

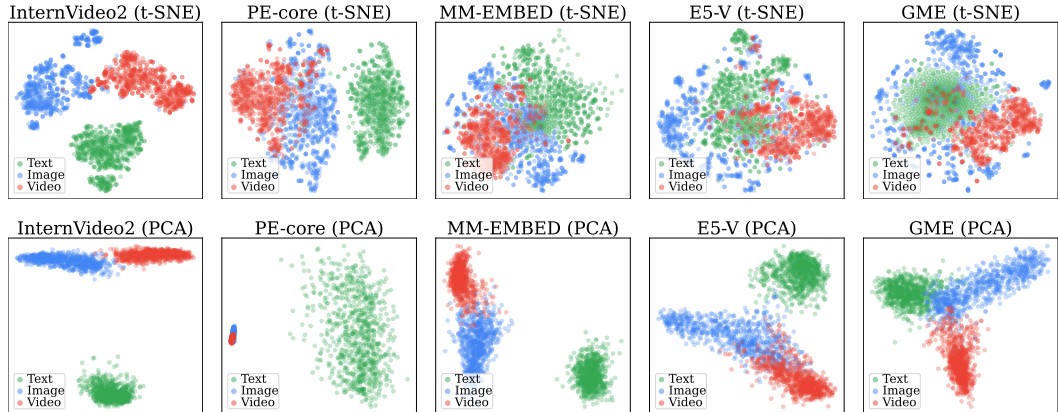

Figure 7: Visualization of the unified embedding space across various multimodal encoders.

# E  MODALITY GAP IN UNIFIED EMBEDDING SPACE

Figure 7 visualizes the modality gap within the unified embedding space of five multimodal encoders (Wang et al., 2024b; Jiang et al., 2024a; Bolya et al., 2025; Lin et al., 2025; Zhang et al., 2025). The PCA plot reveals that embeddings cluster by modality, with text embeddings (shown in green) exhibiting larger distances from those of other modalities. Recent methods like E5-V and GME focus on better aligning these modalities to narrow the gap. However, despite these efforts, a noticeable separation between modalities remains, indicating that current multimodal encoders still struggle to fully unify the embedding space across text, images, and videos. Therefore, the modality routing mechanism of UniversalRAG is required to dynamically direct each query to its corresponding modality-specific embedding space, thereby effectively bridging the modality gap and enhancing retrieval performance.

# F  QUALITATIVE RESULTS

We present case studies to demonstrate the effectiveness of UniversalRAG. Table 12 compares the results of various RAG approaches, including traditional single-modality methods and UniversalRAG, on queries from the WebQA dataset. Traditional approaches such as TextRAG and VideoRAG fail to generate accurate answers: TextRAG retrieves passages lacking relevant visual details, while VideoRAG is better suited for temporal reasoning tasks. In contrast, UniversalRAG correctly routes the query to the image modality, recognizing that visual information about color is necessary, and successfully generates the correct response. This highlights the advantage of modality-aware routing in leveraging the appropriate data from the correct modality corpus, demonstrating UniversalRAG's ability to adaptively select the most informative modality for accurate answer generation.

In addition to modality routing, we observe that UniversalRAG also benefits from retrieving information at the appropriate granularity. Table 13 shows results from HotpotQA, where the query requires complex reasoning over multiple text sources. While paragraph-level granularity fails to provide sufficient context for reasoning, UniversalRAG routes the query to the document-level corpus to

retrieve all the textual information necessary for accurate reasoning. Similarly, for video queries, Table 14 shows results from LVBench on the query that requires only a short segment of the full long video to answer. While full-video-level retrieval includes irrelevant content and uniformly sampled 32 frames fail to capture the necessary information, clip-level retrieval focuses on smaller, more relevant segments of the video to ensure that only the most pertinent visual details are considered, leading to a more accurate answer.

UniversalRAG not only retrieves from the most relevant single modality but also allows cross-modal retrieval, where the router can select more than a single modality-granularity pair when required. Table 15 shows an example from HybridQA, where queries primarily require tables, but can be largely benefit from complementary textual sources. Typically, factual information is best captured from paragraphs, whereas structured knowledge, such as numerical values, is more effectively represented in tables. With its cross-modal retrieval capability, UniversalRAG-Cross successfully retrieves from both modalities, providing the information required to answer the query. In contrast, UniversalRAG-Uni, limited to choose a single modality source, retrieves insufficient evidence to answer correctly.

However, there are some cases where the routing mechanism fails, particularly when the query exhibits ambiguity in modality requirement or when the required information spans across multiple modalities. Table 16 shows failure cases in which UniversalRAG, employing DistilBERT as the trained router, incorrectly routes the modality. The first case shows the case where the router's inductive bias leads it to mistakenly route a query, routing a commonsense query to paragraph-level retrieval. Also, the router sometimes struggles to distinguish between closely related modalities, such as images to short clips that capture specific moments. There are also cases where the router fails to correctly determine granularity, frequently misrouting queries that lie near the boundary between two granularity levels, as shown in the last two examples.

# G   USE OF LARGE LANGUAGE MODELS (LLMs)

We use large language models merely as a writing assistant. Its role is confined to improving grammar and paraphrasing sentences for clarity, and all the core ideas regarding problem definition, UniversalRAG framework, experimental design, and interpretation of results are entirely our own.

Table 12: Case study comparing RAG on single modality against UniversalRAG (Ours). Traditional RAG models are limited by modality-specific retrieval, often missing relevant knowledge from other modalities. UniversalRAG (Ours) dynamically routes each query to the most appropriate modality, enabling more accurate retrieval and response generation.

| **Comparison of UniversalRAG to Single-modality RAG** |
| --- |

| **Question** | What color balloons were displayed on the USNS Carl Brashear during its ship launching ceremony in San Diego? |
| --- | --- |
| **Ground Truth** | The balloons were red, white, and blue that were displayed on the USNS Carl Brashear during the ship launching ceremony in San Diego. |
| **No Retrieval** | The USNS Carl Brashear's ship launching ceremony in San Diego likely featured a diverse range of balloons, but the specific color or colors of the balloons during the ceremony are not universally documented. They may not have been a focus of the event, and such details can vary or not be available in the public record. Therefore, the exact answer to the colors of the balloons on the USNS Carl Brashear may not be specified or may be unknown. ✗ |

**TextRAG**

**Retrieved:** Randall Dark Randall P. Dark is a Canadian writer, director, and producer recognized as an early proponent of HDTV. In 1995, Dark and his company, HD Vision, were involved with the United Nations 50th Anniversary where they shot, in HDTV, President Bill Clinton's speech in San Francisco. In 1999, Dark and HD Vision produced a documentary for PBS about ...

**Response:** The provided documents do not contain any information about the color of the balloons displayed on the USNS Carl Brashear during its ship launching ceremony in San Diego. ✗

**TableRAG**

**Retrieved:**

| Rank | MSA | Region | Chinese-Americans | % |
| --- | --- | --- | --- | --- |
| 1 | New York-Newark-Jersey City | Mid-Atlantic | 739,144 | 3.7 |
| 2 | Los Angeles-Long Beach-Anaheim | Pacific | 528,248 | 4.0 |
| ⋮ | ⋮ | ⋮ | ⋮ | ⋮ |

**Response:** The document provided does not contain any information about the color of the balloons displayed on the USNS Carl Brashear during its ship launching ceremony in San Diego. ✗

**VideoRAG**

**Retrieved:**

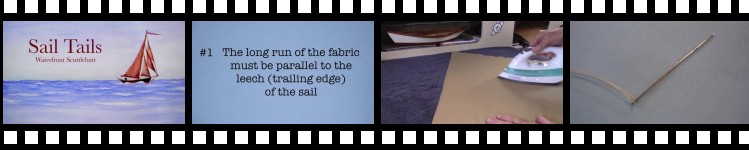

**Response:** Green, blue, and yellow. ✗

**UniversalRAG (Ours)**

**Routed to:** Image

**Retrieved:**

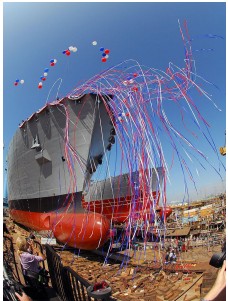

**Response:** The USNS Carl Brashear was adorned with red, white, and blue balloons during its launch ceremony in San Diego. ✓

Table 13: Case study comparing different levels of text granularity. The user query requires complex retrieval involving multiple entities. ParagraphRAG retrieves limited context centered around a single entity, leading to an incorrect answer. UniversalRAG (Ours) routes the query to the document corpus and retrieves richer document-level information, allowing it to capture both relevant entities and generate the correct response.

| Comparison of Using Different Text Granularity | |
|---|---|
| **Question** | George Reed was, along with which Drake University player, one of the greatest running backs in CFL history? |
| **Ground Truth** | Johnny D. Bright |
| **ParagraphRAG** | **Retrieved:** Blue Bombers. He remained in Winnipeg for two seasons (1986-87), but only played in nine games. He finished his career with 4,586 rushing yards, 275 receptions for 2,861 yards and 48 total touchdowns. His son Corey was a sprinter on Clemson's men's track & field team. Lester Brown (Canadian football) Lester Brown (born January 5, 1957) is a former Canadian Football League running back for the Saskatchewan Roughriders, Montreal Concordes, Toronto Argonauts, Ottawa Rough Riders and Winnipeg Blue Bombers. He played college football at Clemson University. Brown attended Myrtle Beach High School, before accepting a football scholarship from Clemson University. 

 **Response:** Lester Brown ✗ |
| **UniversalRAG (Ours)** | **Routed to:** Document 

 **Retrieved:** 
 *George Reed (Canadian football)* George Robert Reed, CM, SOM (born October 2, 1939), is a former American college football and Canadian Football League player. Reed, along with Mike Pringle and Johnny Bright, is one of the players most often mentioned as being the greatest running back in CFL history. In November 2006, Reed was voted one of the CFL's Top 50 players (#2) of the league's modern era by Canadian sports network. 
 ... 
 *Johnny Bright* Johnny D. Bright (June 11, 1930 – December 14, 1983) was a professional Canadian football player in the Canadian Football League. He played college football at Drake University. He is a member of the Canadian Football Hall of Fame, the National Football Foundation's College Football Hall of Fame, the Missouri Valley Conference Hall of Fame, the Edmonton Eskimos Wall of Honour, the Alberta Sports Hall of Fame, and the "Des Moines Register's" Iowa Sports Hall of Fame. 

 **Response:** Johnny Bright ✓ |

Table 14: Case study comparing different levels of video granularity. The user query requires only a segment of the video to determine the answer. VideoRAG retrieves a broad range of frames across the video, which may include irrelevant content or miss key frames, leading to an incorrect response. UniversalRAG (Ours) routes the query to the clip-level granularity, retrieving more focused and relevant visual information, enabling it to generate the correct response.

| Comparison of Using Different Video Granularity | |
| --- | --- |
| **Question** | What does the protagonist observe through the window after being taken to the utility room in the full episode of Blue Eye Samurai on Netflix?
(A) A group of monks sitting cross-legged in the snow
(B) A group of citizens chatting together
(C) A group of warriors practicing swords
(D) A group of samurais eating |
| **Groud Truth** | C |
| **VideoRAG** | **Retrieved:**

(Timestamp Range: 00:00~1:01:05)

**Response:** A ✗ |
| **UniversalRAG (Ours)** | **Routed to:** Clip

**Retrieved:**

(Timestamp Range: 33:46~36:56)

**Response:** C ✓ |

Table 15: Case study comparing UniversalRAG across uni-modal and cross-modal scenarios. In the uni-modal setup, where only a single prominent modality is used, information can sometimes be incomplete as they require evidence across modalities. UniversalRAG-Cross, with its cross-modal capability, gathers evidence from multiple modalities to generate a more comprehensive response.

| Comparison of UniversalRAG across Uni- and Cross-Modal Retrieval | |
|---|---|
| **Question** | What year did an artist known by the mid-1960s in soul and jazz circles for his recording skills release an American record company and label founded by brothers Max and Sol Weiss in 1949? |
| **Ground Truth** | 2000 |
| **UniversalRAG-Uni** | **Routed to:** Paragraph |
| | **Retrieved:** David Axelrod ( April 17 , 1931 [ nb 1 ] - February 5 , 2017 ) was an American composer , arranger , and producer . After starting out as a staff producer for record companies specializing in jazz , Axelrod became known by the mid-1960s in soul and jazz circles for his recording skills . In 1968 , Axelrod embarked on a solo career and released several eccentric albums during the 1970s that showcased his characteristic sound , which combined heavily microphoned drums and baroque orchestration , and avant garde themes ranging from the environment to heightened mental awareness . With his early solo projects , Axelrod was one of the first recording artists to fuse elements of jazz , rock , and R & B . One of his most important records , Song of Innocence ( 1968 ) , featured instrumental interpretations of 18th-century poet William Blake 's poetry collection of the same name done in a contemporary musical vein , leading one critic at the time to coin the term jazz fusion and numerous hip hop producers to sample the album 's music decades later . |
| | **Response:** 1960 ✗ |
| **UniversalRAG-Cross** | **Routed to:** Paragraph+Table |
| | **Retrieved:** (Above Paragraph with the following table) |

| Year | Album | Artist | Genre | Label | Credit |
|---|---|---|---|---|---|
| ⋮ | ⋮ | ⋮ | ⋮ | ⋮ | |
| 1998 | Greatest Hits | Joe Cocker | Rock | EMI Electrola | Trombone on You Can Leave Your Hat On |
| 2000 | The Axelrod Chronicles | David Axelrod | Jazz , funk , soul | Fantasy | Trombone |
| 2004 | Ultimate Collection | Joe Cocker | Rock | Hip-O , A & M | Horn on You Can Leave Your Hat On |
| ⋮ | ⋮ | ⋮ | ⋮ | ⋮ | |

**Response:** 2000 ✓

Table 16: Failure cases in modality routing with UniversalRAG (Ours).

| Question | Ground Truth | UniversalRAG (Ours) |
|---|---|---|
| What is produced during photosynthesis? (A) hydrogen (B) nylon (C) oxygen (D) light | No | Paragraph |
| Who is seated to the right of Kobe in the Jimmy Kimmel tribute show? | Clip | Image |
| What is the name of a type of dual purpose fighter-bomber aircraft used by the US Air Force? | Paragraph | Document |
| What is the main cause of Lee Chong Wei losing points in the first half of his semi-final match against Lin Dan in the Rio 2016 Olympics replay? | Video | Clip |

Classify the following query into one of seven categories: **[No, Paragraph, Document, Table, Image, Clip, Video]**, based on whether it requires retrieval-augmented generation (RAG) and the most appropriate modality. Consider:

- **No**: The query can be answered directly with common knowledge, reasoning, or computation without external data.
- **Paragraph**: The query requires retrieving factual descriptions, straightforward explanations, or concise summaries from a single source.
- **Document**: The query requires multi-hop reasoning, combining information from multiple sources or documents to form a complete answer.
- **Table**: The query requires information that is best represented in a tabular format, often involving comparisons or structured data.
- **Image**: The query focuses on visual aspects like appearances, structures, or spatial relationships.
- **Clip**: The query targets a short, specific moment or event within a video, without needing full context.
- **Video**: The query requires understanding dynamic events, motion, or sequences over time in a video.

**Examples:**
- "What is the capital of France?" → **No**
- "What is the birth date of Alan Turing?" → **Paragraph**
- "Which academic discipline do computer scientist Alan Turing and mathematician John von Neumann have in common?" → **Document**
- "Among the recepients of the Turing Award, who had the earliest birth year?" → **Table**
- "Describe the appearance of a blue whale." → **Image**
- "Describe the moment Messi scored his goal in the 2022 World Cup final." → **Clip**
- "Explain how Messi scored his goal in the 2022 World Cup final." → **Video**
- "Solve $12 \times 8$." → **No**
- "Who played a key role in the development of the iPhone?" → **Paragraph**
- "Which Harvard University graduate played a key role in the development of the iPhone?" → **Document**
- "What is the cheapest iPhone model available in 2023?" → **Table**
- "Describe the structure of the Eiffel Tower." → **Image**
- "Describe the moment Darth Vader reveals he is Luke's father in Star Wars." → **Clip**
- "Analyze the sequence of events leading to the fall of the Empire in Star Wars." → **Video**

Classify the following query: {query}
Provide only the category.

Figure 8: Prompt for query routing in a train-free manner. The prompt defines each category with concise criteria and illustrative examples. Specifically, examples are designed to contrast closely related cases: for example, Paragraph vs. Document for simple fact retrieval vs. multi-hop reasoning; and Clip vs. Video for short specific moments vs. long-term sequential understanding, highlighting the key aspect that differentiates each category.

Evaluate whether the query can be answered using general knowledge about the image's subject rather than relying solely on details unique to the provided image, and verify that the answer is obtainable from the image and the query.
- Respond "yes" if:
    1. The query can be fully answered using general knowledge about the subject.
    2. The answer can be derived solely from the image and the query, without needing image-specific details.
- Respond "no" if either condition is not met.

**Example 1:**
- Image: A portrait of Donald Trump
- Query: What is the color of Trump's hair?
- Answer: White
- Response: "yes"

**Example 2:**
- Image: A close-up photo of a light bulb
- Query: What is the color of the light bulb in this image?
- Answer: Yellow
- Response: "no"

Figure 9: Prompt to filter queries for WebQA.

You will receive a query from a video QA dataset and the title of the corresponding video on YouTube. I want you to paraphrase the query by replacing "in the video?", "of the video", or similar phrases with references to the video content naturally. The output should sound as if a human is asking ChatGPT, and should not explicitly mention the exact name of the video or even parts of the title. However, the rephrased query should contain enough implicit information about the video to allow the model to identify it. Try to reduce the chance of the model getting confused between multiple possible video candidates. If there could be multiple video matches for a given query, try to include more information in the rephrased query.

**Example 1:**
- Query: What year appears in the opening caption of the video?
- Video Title: Blue Eye Samurai | Hammerscale | Full Episode | Netflix
- Upload Date: 2023-11-05
- Channel Name: Netflix
- Rephrased Output: What year appears in the opening caption of the Blue Eye Samurai episode on Netflix?

**Example 2:**
- Query: After the vlogger sees a dog with an advertisement from the company named Smitten, camera changes to the scene with ___.
- Video Title: My ICELAND Experience | Ultimate Travel Vlog
- Upload Date: 2022-10-26
- Channel Name: Kallmekris
- Rephrased Output: After spotting a dog with a Smitten advertisement, what scene does the camera transition to in Kallmekris's Iceland travel vlog from 2022?

Figure 10: Prompt to rephrase queries using video metadata for LVBench and CinePile.

