# OpenReview forum: "UniversalRAG: Retrieval-Augmented Generation over Corpora of Diverse Modalities and Granularities"
_ICLR.cc/2026/Conference — ICLR 2026 Conference Withdrawn Submission_

### Official Review · Reviewer_531v · 2025-10-27

**Soundness:** 2
**Presentation:** 3
**Contribution:** 2
**Rating:** 2
**Confidence:** 4

**Summary:**

The paper introduces UniversalRAG, a retrieval-augmented generation framework designed to handle corpora across multiple modalities (text, image, video, table) and granularities (paragraph, document, clip, video). Instead of a unified multimodal embedding space, the authors propose a modality-aware and granularity-aware routing mechanism that dynamically determines the most suitable corpus for retrieval per query. The method is evaluated across eight datasets spanning different modalities, showing improved accuracy over unimodal or unified embedding-space baselines. Variants include both trained routers and a training-free one.

**Strengths:**

- The paper is clearly written, well structured, and presents the motivation, method, and results in a coherent way. Figures are well-designed and help illustrate the concept.
- The problem motivation is strong and convincing. The authors correctly identify that current universal multimodal retrieval systems have failed to truly unify retrieval across modalities. Existing unified-embedding approaches still often suffer from modality bias, where queries tend to retrieve results from the same modality due to embedding-space clustering. I agree with this diagnosis and find the proposed modality- and granularity-aware routing to be a reasonable and promising alternative to address this limitation.
- The comparison between training-based and training-free (zero-shot) routers is thorough, and the inclusion of analyses such as router accuracy, latency, and scalability is commendable.

**Weaknesses:**

- Although the paper’s goal is to evaluate retrieval and generation across diverse modalities, each benchmark is strictly modality-specific (e.g., all queries in a dataset are either text-, image-, or video-based). This design makes the routing problem trivial, the router effectively learns to identify which dataset corresponds to which modality, not to infer modality from ambiguous, mixed, or cross-modal queries. As a result, the claimed “universal” capability is not meaningfully tested.
- Moreover, a key expectation for a multimodal RAG system is to operate effectively when multiple modalities are present within a single corpus. However, no experiment examines this setting. Without retrieval or routing under multi-modal candidate pools (e.g., setups similar to InfoSeek[1] or E-VQA[2], where text, image, and video evidence coexist), it remains unclear whether the router performs genuine modality reasoning or simply dataset-level selection.
- The benchmark design seems inherently favor UniversalRAG. Because datasets are modality-specific, the routing decision is effectively trivial. This makes the experimental gains less meaningful, as they mainly reflect the advantage of choosing the correct unimodal retriever rather than demonstrating improved generalization or reasoning across modalities. In practice, UniversalRAG’s performance relies too much on the strength of existing modality-specific retrievers.
- The paper omits fair comparisons in terms of model size, memory footprint, and latency across modalities. UniversalRAG combines multiple modality-specific retrievers plus an additional router, which almost certainly increases the total number of parameters and inference cost. Yet, only video-corpus latency is analyzed, with no consistent comparison against other modality-specific or unified baselines. This makes the efficiency analysis incomplete and weakens the fairness of the claimed advantage
- The authors highlight out-of-domain evaluation as an indicator of generalization. However, other baselines are not explicitly evaluated under in-domain vs. out-of-domain separation, making the comparison asymmetric and inherently favorable to UniversalRAG. In the appendix (Table 11), the out-of-domain results show relatively small gains compared to baselines, suggesting that the model’s advantage diminishes outside the curated in-domain setup. In fact, generalization seems to rely heavily on GPT-4.1-based routing; without it, the trained routers (DistilBERT, T5) show clear degradation. This raises concerns about whether the claimed universality stems from the framework or from the use of a powerful external model.
- The authors state that InternVideo2 serves as the vision-specific encoder, yet Table 1 shows discrepancies between VideoRAG results and InternVideo2-based retrieval on the same datasets (VideoRAG-Wiki, VideoRAG-Synth). It is unclear whether these differences arise from implementation variance. This lack of clarity undermines confidence in the experimental reproducibility.

[1] Chen et. al., Can pre-trained vision and language models answer visual information-seeking questions?, EMNLP 2023.
[2] Mensink et. al., Encyclopedic vqa: Visual questions about detailed properties of fine-grained categories. ICCV 2023.

**Questions:**

- The authors argue that current multimodal retrieval suffers from imperfect alignment across modalities. Do the authors view UniversalRAG as a temporary and pragmatic workaround until better multimodal alignment models are achieved, or as a fundamentally necessary paradigm, implying that full alignment across modalities is infeasible and that routing-based retrieval will remain essential in the long term?
- The paper mentions using InternVideo2 as the vision-specific encoder, but Table 1 shows discrepancies between VideoRAG results and InternVideo2-based baselines on the same datasets. Could the authors explain this inconsistency?
- Could the authors provide the total parameter count, memory footprint, and latency of UniversalRAG (router + all retrievers + LVLM) compared to unified baselines such as GME or UniRAG? Without these numbers, fairness and scalability remain unclear.
- I believe the out-of-domain (OOD) evaluation should appear in the main paper, not only in the appendix. The current in-domain results may be biased toward in-domain datasets, so showing OOD performance side-by-side with baselines is essential for fair evaluation.
- Could the authors explain whether the current setup enforces an all-or-nothing assumption, i.e., if the router predicts the wrong modality, the model cannot access any information from that modality? In real-world scenarios, even “incorrect” modalities may contain partially relevant or redundant cues (e.g., a text caption describing a video frame). Does UniversalRAG allow partial credit or soft routing across modalities?
- When the router misclassifies a query’s modality, does the system completely fail to retrieve the correct information? In real-world scenarios, relevant information might exist in other modalities (e.g., a text description of an image).

---

> ### Author Response · Authors · 2025-11-22
> **Response to Reviewer 531v (1/4)**
>
> We sincerely thank you for the constructive feedback and for recognizing the key contributions of our work. We also appreciate your acknowledgement of our novel modality- and granularity-aware routing strategy for addressing the modality gap in existing unified multimodal embedding models, as well as your positive comments on our extensive experiments across comprehensive benchmarks covering multiple modalities and granularities to validate our framework.
>
> ---
> > **W1. Benchmarks are modality-specific, making routing trivial, and leaving the universal capability untested**
>
> We appreciate your concern on the universality claim of UniversalRAG, and we would like to clarify that the "universal" capability refers to UniversalRAG’s ability to dynamically identify and retrieve from the appropriate modality–granularity source within a retrieval pool that contains "any" modality- and granularity-specific corpora. Although each dataset and its corresponding results (each column in `Table 1`) may appear modality-specific when viewed independently, our focus is not to assess performance within individual datasets. Instead, UniversalRAG is evaluated on a comprehensive benchmark that covers a wide range of modality–granularity demands. Therefore, its true universal performance is reflected in the overall results across all datasets, as shown in `Figure 3`.
>
> Furthermore, the routing problem is not always trivial. Even within a single dataset, the correct modality or granularity is not always obvious. For example, WebQA and HybridQA frequently require cross-modal reasoning, making modality selection inherently challenging, while HotpotQA introduces ambiguity in granularity decisions, complicating routing even for strong routers. Beyond individual datasets, our evaluation covers 8 in-domain and 5 out-of-domain benchmarks spanning diverse domains and modality–granularity requirements. This variety introduces substantial routing complexity, and our experiments demonstrate UniversalRAG’s robustness under these challenging settings.
>
> ---
> > **W2. Multimodal RAG with multiple modalities present within a single corpus, and Whether the router performs modality reasoning or dataset-level selection**
>
> We would like to clarify that we already include a baseline that operates in a setting where multiple modalities coexist within a single corpus. The core motivation behind UniversalRAG is to address the limitations of unified-embedding approaches, which project heterogeneous modalities into a single embedding space but fails when multiple modalities coexist in the same corpus due to the inherent modality gap. Our experimental results, particularly weak performance of unified-embedding baselines in `Table 1` and their text-biased retrieval behavior shown in `Figure 4`, clearly demonstrate the limitations of unified embedding approaches where multiple modalities are projected to a single corpus. To address these issues, UniversalRAG introduces a modality-aware routing by directing each query to the most appropriate modality-specific corpus. As shown in `Table 1`, UniversalRAG consistently outperforms baseline methods that rely on unified embeddings, demonstrating effectiveness of routing-based retrieval in complex multimodal pools.
>
> Moreover, UniversalRAG’s router is designed and evaluated to infer the modality requirement of each query, rather than performing dataset-level selection. To test this capability, we constructed a comprehensive retrieval pool by aggregating multiple datasets across different modalities and granularities, thereby forming multiple modality- and granularity-specific corpora. Given the presence of ambiguous queries and multiple datasets within each modality–granularity pair, our evaluation demonstrates that the router goes beyond simple dataset selection and instead performs genuine modality reasoning.

---

> > ### Author Response · Authors · 2025-11-22
> > **Response to Reviewer 531v (2/4)**
> >
> > > **W3. Benchmark design inherently favors UniversalRAG and relies on the strength of modality-specific retrievers**
> >
> > We appreciate your concern about our experimental setting, particularly regarding the datasets. However, we would like to emphasize that our benchmark design does not inherently favor UniversalRAG. As discussed in `W1`, our evaluation is conducted on a unified corpus constructed from diverse benchmarks spanning multiple modalities, granularities, and domains. Under this unified setup, the routing problem is far from trivial. Our comprehensive evaluation demonstrates that UniversalRAG’s modality-aware routing performs robustly across all these heterogeneous settings, indicating genuine reasoning capability across modalities and granularities, rather than simply selecting a unimodal retriever.
> >
> > Moreover, the ability to leverage modality-specific retrievers is a design advantage of UniversalRAG. Our flexible framework enables seamless integration of any state-of-the-art modality-specific retriever, allowing the system to fully exploit the strengths of specialized models without relying on a unified embedding space, which is highly challenging to align across heterogeneous modalities. `Table D.1` shows the performance of UniversalRAG by changing each text and visual retriever from the current bge-large-en-v1.5 and InternVideo2 to Qwen3-Embedding-4B and VLM2Vec-V2.
> >
> > **Table D.1. Performance of UniversalRAG (DistilBERT) with different retriever choices**
> > |Retriever|MMLU (Acc)|NQ (F1)|HotpotQA (F1)|HybridQA (F1)|WebQA (R-L)|LVBench (Acc)|VideoRAG-Wiki(R-L)|VideoRAG-Synth(R-L)|Avg.|
> > |-|-|-|-|-|-|-|-|-|-|
> > |Ours|62.50|47.08|26.96|12.04|46.32|35.65|19.23|28.23|36.82|
> > |Qwen3-Embedding-4B|**63.00**|**47.86**|27.75|**14.86**|46.32|35.26|19.23|28.23|37.61|
> > |VLM2Vec-V2|62.50|47.08|26.63|12.00|**47.13**|36.04|**20.05**|29.75|37.33|
> > |Qwen3-Embedding-4B + VLM2Vec-V2|**63.00**|**47.86**|**27.85**|**14.86**|**47.13**|**36.14**|**20.05**|**29.77**|**38.37**|
> >
> > The results show that incorporating stronger modality-specific retrievers consistently leads to additional performance gains, demonstrating UniversalRAG's strong capability to integrate and benefit from any modality-specific retrievers. With its flexible framework, it can be also seamlessly be extended to new modalities by adapting corresponding corpus and modality-specific retriever.
> >
> > ---
> > > **W6, Q2. Clarification on InternVideo2 as visual encoder vs. unified embedding baseline**
> >
> > We apologize for any confusion. This appears to stem from a misunderstanding, and we would like to clarify the two distinct roles in which InternVideo2 is used in our paper. While we use InternVideo2 as a visual encoder for UniversalRAG, as described in the "Implementation Details" section of `Section 3.1`, we also introduce "InternVideo2" as a unified embedding baseline in the “Methods” section, where InternVideo2 is employed as a multimodal encoder to embed all data modalities into a unified embedding space. Throughout the paper, except in the "Implementation Details" section where we describe the encoder itself, "InternVideo2" is used to denote this unified embedding baseline method rather than the visual encoder itself. We apologize if this may have caused confusion, and to prevent further confusion, we have added a footnote to clarify the differences of these two identical terms.
> >
> > Regarding the discrepancy noted between VideoRAG and InternVideo2 in `Table 1`, VideoRAG is a unimodal baseline that retrieves only from the video corpus, using InternVideo2 as a video encoder. In contrast, the InternVideo2 baseline is a unified embedding method that encodes all items across multiple modalities into a shared embedding space. As shown in `Figure 4`, the resulting modality gap causes the InternVideo2 baseline to retrieve textual items for every query. Consequently, although both methods rely on InternVideo2 as the underlying encoder, VideoRAG retrieves exclusively from videos, whereas the InternVideo2 baseline tends to retrieve text from the multimodal corpus. This fundamental difference leads to the large performance gap observed on the VideoRAG datasets.

---

> > > ### Author Response · Authors · 2025-11-22
> > > **Response to Reviewer 531v (3/4)**
> > >
> > > > **W4, Q3. Efficiency analysis against baseline methods**
> > >
> > > Thank you for your question regarding retrieval efficiency. The comparison between UniversalRAG and the unified embedding baseline is already presented in "Retrieval Efficiency of Modality-Specific Retrieval" of `Section 3.2`. `Figure 5` shows retrieval latency for both UniversalRAG and the unified embedding baseline (InternVideo2; as a unified embedding method, not a video-specific retriever as clarified in `W6, Q2`) across varying corpus sizes. UniversalRAG’s modality-specific retrieval enables the retriever to search only within the targeted modality-specific corpus rather than over the entire multimodal embedding space used in unified embedding baselines. This substantially reduces the effective search space, leading to sub-linear retrieval latency, while unified embedding baselines exhibit linear latency, as the routing overhead in UniversalRAG is fixed and small compared to the cost of search.
> > >
> > > Also, regarding memory footprint, UniversalRAG introduces an additional increase in need for GPU memory compared to unified embedding methods. While both cases require every item in the corpus to be encoded using either a modality-specific or a multimodal encoder, UniversalRAG requires the simultaneous loading of all modality-specific encoder models to achieve its low retrieval latency. Meanwhile, despite this increased background memory requirements, UniversalRAG provides superior retrieval latency and improved performance, making it a promising choice for real-world applications where speed and accuracy are prioritized.
> > >
> > > ---
> > > > **W5, Q4. OOD performance and reliance on powerful router**
> > >
> > > We deeply appreciate your concern. Our original discussion on OOD generalization in the main text was limited to comparing UniversalRAG variants in `Table 6`, leaving the detailed result with all other baselines to the Appendix due to the page limit. With the extended limit, we have revised the manuscript to integrate this comprehensive evaluation into the main body. We have relocated the full OOD performance table (previously `Table 11` of the Appendix) into the main text (now `Table 4`). Furthermore, we have added an explicit discussion on the symmetric comparison of in- and out-of-domain results.
> > >
> > > The results show that training-free router exhibits strong performance on OOD datasets, maintaining our framework's generalization capability. While trained routers show degradation due to truly unseen distributions, especially for the domain shift of LaRA, our ensemble-based routing offers a promising solution for achieving robust generalization in both in-domain and out-of-domain settings.
> > >
> > > Finally, although stronger external models can be beneficial, UniversalRAG can achieve comparable results with even smaller models. `Table D.2` shows results using Qwen3-4B-Instruct, a much smaller model than GPT-4.1, as a training-free router.
> > >
> > > **Table D.2. Performance of UniversalRAG with Qwen3-4B-Instruct**
> > > ||MMLU (Acc)|NQ (F1)|HotpotQA (F1)|HybridQA (F1)|WebQA (R-L)|LVBench (Acc)|VideoRAG-Wiki(R-L)|VideoRAG-Synth(R-L)|Avg.|
> > > |-|-|-|-|-|-|-|-|-|-|
> > > |UniversalRAG (GPT-4.1)|**65.00**|**47.60**|23.84|**12.13**|44.74|31.15|13.95|22.49|35.27|
> > > |UniversalRAG (Qwen3-4B-Instruct)|**65.00**|47.41|**24.11**|11.14|**45.03**|**31.54**|**14.82**|**24.35**|**36.02**|
> > >
> > > These results show that using Qwen3-4B, a much smaller model than GPT-4.1, achieves comparable or even better average performance, demonstrating that the universality of our framework stems from the routing mechanism, not the intrinsic power of the external models employed.

---

> > > > ### Author Response · Authors · 2025-11-22
> > > > **Response to Reviewer 531v (4/4)**
> > > >
> > > > > **Q1. UniversalRAG as a fundamentally necessary approach**
> > > >
> > > > Rather than considering UniversalRAG as a temporary workaround, we believe UniversalRAG is a fundamentally essential paradigm for multimodal RAG. Constructing a perfectly aligned unified embedding space across heterogeneous modalities is extremely challenging, and becomes increasingly unrealistic as new modalities are introduced. UniversalRAG directly addresses these challenges through modality-aware routing, which effectively mitigates modality gaps and and retrieves from the most appropriate modality-specific corpus. This design makes UniversalRAG a practical, extensible, and future-proof framework for handling diverse modalities and granularities, especially in realistic scenarios where multiple modalities coexist within the same corpus.
> > > >
> > > > This observation aligns with how large-scale search systems operate in practice. Even platforms such as Google Search surface separate retrieval results for different modalities (e.g., "Images", "Videos"), reflecting the inherent difficulty of aligning heterogeneous information sources into a single unified corpus. Therefore, given the complexity and diversity of modern multimodal data, we believe that UniversalRAG's modality-aware routing is not merely an alternative, but a fundamentally necessary approach for effective multimodal RAG.
> > > >
> > > > ---
> > > > > **Q5, Q6. All-or-nothing assumption and partial credit from incorrect routing**
> > > >
> > > > Existing unimodal or unified embedding approaches often fail when queries require evidence from different modalities, largely due to the inherent modality gap, leading to a "nothing-and-nothing" outcome when the retrieval is mismatched. In contrast, UniversalRAG’s routing mechanism produces an "all-or-nothing" behavior in which correct routing yields correct outcome while incorrect routing may provide no useful evidence.
> > > >
> > > > Meanwhile, even when routed to an imperfect modality, our approach can still obtain partial credit. For example, a query such as "What is the method for making vanilla extract" in the VideoRAG-Wiki is best supported by video evidence because the answer requires visually grounded procedural details. However, our model can still retrieve meaningful evidence from textual evidence within the paragraph corpus, allowing it to retrieve partial procedural details even without access to the video.
> > > >
> > > > Moreover, our proposed cross-modal variant supports soft routing that retrieves information from multiple modality- and granularity-specific corpora when a query is ambiguous or inherently requires cross-modal references. The combination of partial credit and this routing flexibility shifts the system from strictly all-or-nothing to a more robust "all-or-something." Additional techniques such as cross-modal reranking can further complement the router and strengthen system reliability, which we leave as an interesting future work direction.

---

### Official Review · Reviewer_AeFC · 2025-10-30

**Soundness:** 3
**Presentation:** 3
**Contribution:** 3
**Rating:** 6
**Confidence:** 5

**Summary:**

The paper replaces the common practice of "retrieving in a unified embedding space" with the execution trajectory of "querying Self-Adaptation routing to modality-granularity-corpus", directly addressing a pain point in real systems (different problems require evidence from different sources and units). From the perspective of experimental coverage and analysis accuracy, the paper is of high quality. The novelty is mainly reflected in unifying "modality selection", "granularity selection", and "non-retrieval" into routing decisions, and validating the implementation of this paradigm in real settings of multi-granularity/multi-corpus.

**Strengths:**

1. Unifying "Modal Selection", "Granularity Selection", and "Non-Retrieval" into query-level routing decisions and systematically validating them in real-world environments of MultiModal Machine Learning/multi-granularity/multi-corpus is a powerful alternative to the existing "Unified Embedding" paradigm. Compared with the multi-modal RAG approach centered on "unified space + unified metric", emphasizing a workflow centered on "query-task requirements" is innovative at both the methodological and systematic levels.

2. Helps to promote the engineering implementation of Multimodal RAG: Compared with the unified embedding paradigm, it is easier to integrate new modalities (by extending routing logic and dedicated retrievers rather than recalibrating the shared space).

**Weaknesses:**

1. Route Cost and System Throughput: Under industrial-scale (>100M entries) and high concurrency, what are the total latency and throughput of route invocation + dedicated retrieval? Can the router be distilled into a smaller model or cached to further reduce online overhead?

2. Learning objective of granularity selection: Has there been an attempt to bind "retrieval granularity" with "generation quality/cost" to a joint objective (e.g., learning with metric-cost Pareto optimality as a constraint), so that the router explicitly weighs the cost during inference?

3. Evidence Aggregation and Conflict: When cross-modal evidence conflicts (image and text are inconsistent), how does the current system make a decision?

**Questions:**

Please refer to Weaknesses.

---

> ### Author Response · Authors · 2025-11-22
> **Response to Reviewer AeFC (1/2)**
>
> We sincerely appreciate your thoughtful review and your recognition of the key contributions of our work. We are pleased that you highlighted the UniversalRAG's novelty and effectiveness of query-level routing for modality and granularity selection, and its flexibility in seamlessly incorporating additional modalities and corresponding retrievers, supporting a truly universal multimodal RAG framework. We carefully considered all of your comments and concerns, and have made every effort to address them thoroughly.
>
> ---
> > **W1. Route cost and system throughput**
>
> As discussed in `Section 3.2` "Retrieval Efficiency of Modality-Specific Retrieval," UniversalRAG serves as an efficient solution to unified-embedding RAG approaches. By performing modality- and granularity-aware routing, the retriever searches only within a targeted subset of the corpus rather than the full embedding space. This reduces the effective search space, yielding sub-linear retrieval latency, in contrast to the linear growth observed in unified retrieval methods over the entire corpus. As shown in `Figure 5`, our approach achieves lower latency than a unified-embedding approach on our 2.63M-scale experimental setting, and this gap is expected to widen as the corpus size increases.
>
> We further analyze how this trend extends to industry-scale corpora (100M). According to our empirical analysis, DistilBERT-based router requires only $t_\text{route}=7.03\text{ ms}$ per query, while similarity search costs $t_\text{search/1M}=6.90\text{ ms}$ per 1M items. Under a uniform modality–granularity distribution across seven corpus partitions, the expected per-query latency of UniversalRAG is
> $$t_\text{route}+\frac{1}{7}\times100\ t_\text{search/1M}\approx105.60\text{ ms},$$
> whereas a unified-embedding approach requires
> $$100\ t_\text{search/1M}\approx690\text{ ms}.$$
> This represents a substantial latency reduction, demonstrating that our routing-based approach scales more favorably than unified retrieval as corpus size grows.
>
> Furthermore, model distillation can offer additional efficiency gains, as smaller routing models still maintain competitive performance, as illustrated in `Figure 6`. In particular, DistilBERT provides a balanced trade-off, showing performance comparable to larger models while benefiting from substantially lower cost due to its compact size (66M). In real-world deployments, the routing overhead can be reduced even further through instruction caching, since a fixed few-shot instruction is reused across queries. Together, these make UniversalRAG a practical and scalable retrieval strategy, especially in large-scale real-world applications.
>
> ---
> > **W2. Learning objective of granularity selection**
>
> Thank you for suggesting this direction. In our current work, we do not explicitly optimize retrieval granularity together with generation cost as part of the training objective; our method focuses on designing an accurate modality- and granularity-aware routing strategy that selects the appropriate retrieval configuration for each query. However, we agree that incorporating a joint cost–quality objective, potentially through a reinforcement learning framework, could enable the model to balance the trade-off between quality and cost during inference, and we consider this an interesting avenue for future work.

---

> > ### Author Response · Authors · 2025-11-22
> > **Response to Reviewer AeFC (2/2)**
> >
> > > **W3. Evidence aggregation and conflict**
> >
> > As you pointed out, cross-modal retrieval can sometimes lead to conflicts between evidence retrieved from different modalities. To understand how current system handles such conflicts, we conduct an experiment in which inference is run using only a single modality, either paragraph or image, for queries where at least one modality is retrieved correctly by UniversalRAG (Cross-GPT-4.1). Based on the retrieval outputs on WebQA, we group queries by whether the retrieved paragraph, the retrieved image, or both are correct. For each group, we compare inference performance under paragraph-only evidence, image-only evidence, and the full UniversalRAG setting where both modalities are provided.
> >
> > **Table C.1. Inference performance by retrieved modality condition**
> > |Retrieved condition|ParagraphRAG|ImageRAG|UniversalRAG (Ours)|
> > |-|-|-|-|
> > |Paragraph only correct|42.72|39.53|41.20|
> > |Image only correct|35.48|51.97|49.58|
> > |Both modalities correct|38.70|47.33|49.13|
> >
> > *(metric: ROUGE-L)*
> >
> > `Table C.1` shows that when correct evidence is present in only one modality, restricting inference to that modality gives the strongest performance. UniversalRAG remains highly robust even when additional incorrect evidence from another modality is provided, and its performance remains close to the corresponding single modality inference. This demonstrates that UniversalRAG can effectively down-weight or ignore uninformative cross-modal evidence and show robust inference quality even in the presense of conflicting information.

---

> > > ### Comment · Reviewer_AeFC · 2025-11-23
> > >
> > > Thank you for your responses. I've decided to maintain my ratings.

---

### Official Review · Reviewer_hhJP · 2025-10-31

**Soundness:** 2
**Presentation:** 2
**Contribution:** 2
**Rating:** 4
**Confidence:** 2

**Summary:**

The paper proposes UniversalRAG to handle diverse multi-modal knowledge corpora and multiple retrieval granularities (e.g., paragraph vs. document, clip vs. full video). The core insight is that existing multimodal RAG systems suffer from a modality gap: queries (typically textual) are biased toward retrieving items of the same modality, even when other modalities contain more relevant information. UniversalRAG introduces a modality- and granularity-aware routing mechanism that dynamically selects the most appropriate corpus (or corpora) for each query, which is training-free manner.

**Strengths:**

1. The paper makes a conceptually clean and timely contribution by reframing multimodal RAG as a routing problem rather than a unified embedding problem
2. The experimental design is comprehensive, constructing a multi-modal, multi-granularity benchmark suite that covers realistic query types (factoid, multi-hop, visual, temporal)

**Weaknesses:**

1. The performance of UniversalRAG hinges critically on the router’s accuracy. While the paper explores multiple router implementations, it does not sufficiently address failure modes in ambiguous or cross-modal queries, as shown in Table 16.
2. Although the paper includes a “Cross-GPT-4.1” variant that retrieves from multiple modalities, the majority of benchmarks are unimodal by design (e.g., NQ is text-only, WebQA is image-only). The true value of cross-modal retrieval is only demonstrated on HybridQA and WebQA subsets. A dedicated benchmark with intrinsically multimodal queries (e.g., “Compare the architectural style shown in this image with the historical description in this paragraph and the example video”) would better validate the framework’s full potential.
3.Training-free router limitations: While GPT-4.1 shows strong out-of-domain generalization, its use as a router raises concerns about cost, latency, and reproducibility (as acknowledged in the Reproducibility Statement).

**Questions:**

Some highlighted performance scores in Table 1 are problematic: (1) NQ F1 (47.89 of ParagraphRAG > 47.86 of UniversalRAG (Cross-GPT-4.1)); (2) HoptpotQA F1 (28.49 of UniversalRAG (Cross-GPT-4.1) > 27.56 of UniversalRAG (T5-Large).

---

> ### Author Response · Authors · 2025-11-22
> **Response to Reviewer hhJP (1/2)**
>
> Thank you very much for taking the time to review our paper. We appreciate your recognition of our novel contribution to multimodal RAG through modality-aware routing for mitigating the modality gap, as well as your acknowledgment of our comprehensive experiments across datasets spanning diverse modalities and levels of granularity.
>
> > **W1. Dependence of performance on router's accuracy and addressing failure modes**
>
> We appreciate your concern. However, as shown in Figure 6, the router already achieves consistently high accuracy across diverse model families, suggesting that routing reliability is not a major limitation in practice. Moreover, while building a perfect router that always assigns each query to the optimal modality and granularity is valuable, the primary focus of our work is to introduce modality-aware routing as a promising direction for multimodal RAG, particularly as a way to mitigate the modality gap inherent in existing unified embedding approaches.
>
> Furthermore, the failure cases in `Table 16` represent only a small subset of scenarios where routing does not perfectly match the ground truth. Even in these mis-routed situations, however, our system often receives partial credit because retrieved information can still be useful. For example, the query "What is produced during photosynthesis?" is labeled as requiring no retrieval, yet retrieving from textual sources can yield even more accurate grounded answer. Likewise, the query "What is the method for making vanilla extract" in the VideoRAG-Wiki is best supported by visually grounded procedural content, but textual evidence remains informative, allowing it to provide partial procedural details even without video.
>
>
> Importantly, for queries that are ambiguous or inherently cross-modal, we already include a cross-modal variant of UniversalRAG that performs soft routing and retrieves from multiple modality- and granularity-specific corpora when needed. As shown in `Table 1`, UniversalRAG (Cross-GPT-4.1) achieves strong performance on HybridQA and WebQA, demonstrating that ambiguous or cross-modal queries benefit from heterogeneous retrieval across modalities. These results collectively show that our current routing strategy is already strong and remains robust even when routing is imperfect, while further improving router accuracy would be an interesting direction for future work.
>
>
> ---
> > **W2. Cross-modal performance underrepresented and evaluations on intrinsic multimodal queries**
>
> Thank you for raising this point. However, this concern stems from the current availability of benchmarks rather than from UniversalRAG itself, as most existing RAG datasets are designed such that each query can be answered using a single dominant modality. Consequently, opportunities to evaluate true multi-source cross-modal reasoning remain limited. To address this, we included HybridQA and the cross-modal subsets of WebQA, which are among the few benchmarks that explicitly require integrating information across modalities.
>
> Furthermore, although our primary evaluation focuses on text-form queries, we additionally conducted the suggested experiments on InfoSeek [1] and MRAG-Bench [2], which contain multimodal queries requiring joint retrieval across modalities. As shown in `Table B.1`, UniversalRAG achieves the strongest performance across both benchmarks, outperforming all baseline retrievers. Furthermore, the cross-modal variant, UniversalRAG (Cross-GPT-4.1), consistently surpasses the unimodal version, achieving improvements of 2.88% on InfoSeek and 2.78% on MRAG-Bench. These results highlight UniversalRAG’s ability to effectively leverage cross-modal knowledge, even in scenarios where queries are inherently multimodal.
>
> **Table B.1. Evaluation of UniversalRAG and baselines on InfoSeek and MRAG-Bench**
> |Retriever|InfoSeek (Acc)|MRAG-Bench (Acc)|
> |-|-|-|
> |Naive|21.68|43.68|
> |ParagraphRAG|44.89|44.86|
> |DocumentRAG|35.70|42.72|
> |TableRAG|17.94|37.77|
> |ImageRAG|32.39|51.22|
> |ClipRAG|18.25|37.03|
> |VideoRAG|18.86|36.73|
> |InternVideo2|41.70|43.46|
> |GME|42.25|45.16|
> |All|32.88|45.01|
> |-|
> |UniversalRAG (GPT-4.1)|43.11|49.67|
> |UniversalRAG (Cross-GPT-4.1)|**45.99**|**52.11**|

---

> > ### Author Response · Authors · 2025-11-22
> > **Response to Reviewer hhJP (2/2)**
> >
> > > **W3. Training-free router limitations**
> >
> > The training-free router demonstrates strong out-of-domain performance, though it may incur higher cost and latency compared with small trained routers, as you pointed. While a powerful model can function effectively as a training-free router, UniversalRAG can achieve comparable results even when using smaller, more efficient models. `Table B.2` presents results using Qwen3-4B-Instruct as the training-free router.
> >
> > **Table B.2. Performance of UniversalRAG with Qwen3-4B-Instruct**
> > ||MMLU (Acc)|NQ (F1)|HotpotQA (F1)|HybridQA (F1)|WebQA (R-L)|LVBench (Acc)|VideoRAG-Wiki(R-L)|VideoRAG-Synth(R-L)|Avg.|
> > |-|-|-|-|-|-|-|-|-|-|
> > |UniversalRAG (GPT-4.1)|**65.00**|**47.60**|23.84|**12.13**|44.74|31.15|13.95|22.49|35.27|
> > |UniversalRAG (Qwen3-4B-Instruct)|**65.00**|47.41|**24.11**|11.14|**45.03**|**31.54**|**14.82**|**24.35**|**36.02**|
> >
> > These results show that UniversalRAG, even when paired with a much smaller 4B model rather than GPT-4.1, achieves comparable or even better performance. This demonstrates that the UniversalRAG framework generalizes effectively while offering lower cost, reduced latency, and improved reproducibility, without dependence on proprietary models.
> >
> > ---
> > > **Q1. Highlight error in `Table 1`**
> >
> > Thank you very much for your careful reading of our main results. We appreciate the opportunity to clarify the issues you raised regarding the highlighted scores in `Table 1`.
> > 1. NQ F1 (47.89 of ParagraphRAG > 47.86 of UniversalRAG (Cross-GPT-4.1))
> >
> > As stated in `Table 1`’s caption, we highlight only the UniversalRAG variants. Since most datasets have ground-truth modality requirements, the corresponding uni-modal, modality-specific retriever serves as an oracle method; therefore, highlighting those values would be misleading and uninformative. Instead, we focus the highlighting exclusively on UniversalRAG variants so that the effects of different router designs of UniversalRAG can be compared.
> >
> > 2. HotpotQA F1 (28.49 of UniversalRAG (Cross-GPT-4.1) > 27.56 of UniversalRAG (T5-Large))
> >
> > The value 28.49 corresponds to the Oracle row, not to UniversalRAG (Cross-GPT-4.1). The actual F1 score of UniversalRAG (Cross-GPT-4.1), shown directly above the Oracle row, is 24.21, which is indeed lower than 27.56 of UniversalRAG (T5-Large).
> >
> > ---
> > [1] Can Pre-trained Vision and Language Models Answer Visual Information-Seeking Questions?, EMNLP 2023.
> >
> > [2] MRAG-Bench: Vision-Centric Evaluation for Retrieval-Augmented Multimodal Models, ICLR 2025.

---

> > > ### Comment · Reviewer_hhJP · 2025-11-23
> > >
> > > It's truly impressive that the qwen3-4b-instruct model outperforms GPT-4.1—I really enjoyed your extended experiment. Could you please further elaborate on the experimental setup? This would help clarify the core reasons behind the current performance differences among models on this router task. For example, could reasoning-focused models like DeepSeek-R1 perform even better?

---

> ### Author Response · Authors · 2025-11-26
>
> Thank you for your follow-up comments, and we're pleased that you enjoyed our additional experiments.
>
> > **Could you please further elaborate on the experimental setup?**
>
> For the experiments in `Table B.2`, we employ both GPT-4.1 and Qwen3-4B-Instruct as training-free router models for UniversalRAG. Both routers are prompted with the same instruction prompt, which includes detailed routing guidelines covering diverse modalities and granularities, along with few-shot examples, as shown in `Figure 8` of Appendix. The response generation model, which is different from the router model, is fixed to InternVL2.5-8B. In short, the only difference between UniversalRAG (GPT-4.1) and UniversalRAG (Qwen3-4B-Instruct) is the choice of router model.
>
> ---
> > **Could reasoning-focused models like DeepSeek-R1 perform even better?**
>
> We appreciate your question regarding the use of reasoning-based routers instead of instruction-tuned (i.e., non-reasoning) router models. To further examine this, we experiment UniversalRAG with open-source reasoning models and compare them against their instruction-tuned variants, with all routers used in a training-free manner and given the same instruction prompt. `Table B.3` shows full results comparing UniversalRAG's performance using reasoning and non-reasoning routers.
>
> **Table B.3. Performance of UniversalRAG comparing reasoning and non-reasoning routers**
>
> *(Each column highlights the best model other than GPT-4.1 in **bold**.)*
> |Router Model|MMLU (Acc)|NQ (F1)|HotpotQA (F1)|HybridQA (F1)|WebQA (R-L)|LVBench (Acc)|VideoRAG-Wiki(R-L)|VideoRAG-Synth(R-L)|Avg.|
> |-|-|-|-|-|-|-|-|-|-|
> |GPT-4.1|65.00|47.60|23.84|12.13|44.74|31.15|13.95|22.49|35.27|
> |-|
> |DeepSeek-R1-Distil-Qwen-1.5B|64.00|43.86|19.90|11.00|43.27|30.53|14.11|22.09|34.21|
> |Qwen3-4B-Instruct|**65.00**|**47.41**|**24.11**|11.14|**45.03**|**31.54**|**14.82**|**24.35**|**36.02**|
> |Qwen3-4B-Thinking|64.50|44.57|22.87|**11.43**|42.21|30.46|14.22|23.81|34.89|
> |Phi-4-mini-instruct|63.00|43.14|23.06|11.00|43.04|31.15|14.19|22.71|34.63|
> |Phi-4-mini-flash-reasoning|63.50|44.00|24.59|11.14|42.77|**31.54**|14.02|22.34|35.02|
>
> Our results demonstrate that reasoning routers show only marginal performance differences compared to non-reasoning routers because effective routing in UniversalRAG primarily depends on identifying the query’s modality and information needs rather than performing extended multi-step reasoning, which is the capability that reasoning models are specifically trained to provide. As a result, even on datasets such as HotpotQA and WebQA, where routing decisions are further complicated by granularity and cross-modality ambiguity, reasoning routers do not demonstrate clear superiority.
>
> Moreover, reasoning-based routers introduce substantial inefficiency. Instead of generating a single-token routing decision, they generate long reasoning chains, resulting Qwen3-4B-Thinking to take about 31 seconds per query, compared with 74ms for Qwen3-4B-Instruct. Given the marginal performance gains and the significant latency overhead introduced by reasoning routers, non-reasoning routers remain the more effective and efficient practical choice for UniversalRAG.
>
> ---
>
> Once again, we appreciate your time and effort in reviewing our work. We sincerely hope this response fully address your questions and would be happy to discuss any additional points.

---

### Official Review · Reviewer_Vohn · 2025-11-01

**Soundness:** 2
**Presentation:** 3
**Contribution:** 2
**Rating:** 4
**Confidence:** 1

**Summary:**

The paper targets the “modality gap” that arises when all items are embedded into one unified space, showing that text-typed queries bias retrieval toward text even when image or video evidence is needed.It proposes a router that first picks modality, then granularity (paragraph vs document; clip vs full video; plus a no-retrieval path), and retrieves only within the chosen corpora using modality-specific encoders.

**Strengths:**

1. Modality+granularity routing is well specified, including a training-based router and a training-free LLM-prompted router, and the formulation includes a clear “no retrieval” option.

2. The modality-gap diagnosis is concrete and visually supported, which justifies routing instead of naive unification.

**Weaknesses:**

1. The “automatic” label construction for router training maps datasets to routing targets, which risks encoding benchmark priors into the router and inflating in-domain gains.

2. The evaluation relies on specific retrievers per modality (e.g., bge-large for text and InternVideo2 for vision) without sensitivity analyses to retriever choice or corpus indexing settings.

3. Cross-modal scenarios are underrepresented in common benchmarks, and even the paper notes stronger gains where queries genuinely require multi-source aggregation.

**Questions:**

see weakness

---

> ### Author Response · Authors · 2025-11-22
> **Response to Reviewer Vohn (1/2)**
>
> Thank you very much for taking the time to review our paper and for providing such thoughtful feedback. We sincerely appreciate your recognition of our analysis of the modality-gap issues in current multimodal RAG systems, as well as our proposed UniversalRAG framework with its novel modality-aware routing mechanism, which selects and retrieves from diverse modality–granularity options, including a no-retrieval path.
>
> ---
> > **W1. Automatic label construction and benchmark prior**
>
> Thank you for raising this issue. Since there is no large-scale dataset that provides query-level annotations specifying the required modalities, we leverage dataset characteristics to automatically construct training labels for the routers in UniversalRAG. While our benchmarks are carefully curated to match modality requirements, we agree that this approach may introduce inductive biases from benchmark priors. To explore the extent of this bias, we conduct an analysis comparing our inductively constructed labels with independently generated silver labels. Specifically, we randomly sampled 100 queries from each inductive-bias category and used GPT-4.1-mini to evaluate each query across all modality–granularity pairs, selecting the best-performing option as the silver label.
>
> **Table A.1. Correlation of inductive biases with silver-label annotations**
> ||None|Paragraph|Document|Table|Image|Clip|Video|Avg.|
> |-|-|-|-|-|-|-|-|-|
> |Match Rate (%)|95|100|100|92|100|66|100|93|
>
> `Table A.1` shows strong alignment for most modalities. While the Clip modality exhibits a lower match rate, the mismatches largely occur between Clip and Video, reflecting their semantic similarity and overlapping content coverage. Overall, these results indicate that benchmark priors serve as reasonably reliable labels for router training.
>
> We also acknowledge that such labels may favor in-domain evaluation. To assess generalization, we further tested UniversalRAG on out-of-domain datasets, as detailed in `Table 4`. Across these evaluations, UniversalRAG achieves strong performance on several datasets, indicating that the router can generalize beyond the benchmarks used during label construction. Nonetheless, we observe failures when the query domains differ substantially from those seen in training. These limitations stem not from the automatic label-construction method itself, but from the limited diversity and scale of available multimodal benchmarks. Expanding coverage, either by curating datasets spanning more domains, modalities, and granularities or by generating additional training data and applying reinforcement learning, offers a promising direction, while it remains beyond the scope of this study.
>
> ---
> > **W2. Sensitivity analyses to retriever choice or corpus indexing settings**
>
> UniversalRAG has a modular design that allows it to flexibly incorporate any modality-specific encoder. Therefore, it can seamlessly integrate any state-of-the-art modality-specific encoders to improve retrieval and overall performance. To understand how corpus encoding influences system effectiveness, we conduct sensitivity analyses using multiple encoder configurations. Specifically, we evaluate two recent state-of-the-art encoders, Qwen-Embedding-4B for text and VLM2Vec-V2 for vision, as well as two classical encoders, DPR and CLIP. `Table A.2` compares these against our current UniversalRAG configuration.
>
> **Table A.2. Performance of UniversalRAG (DistilBERT) with different retriever choices**
> |Retriever|MMLU (Acc)|NQ (F1)|HotpotQA (F1)|HybridQA (F1)|WebQA (R-L)|LVBench (Acc)|VideoRAG-Wiki(R-L)|VideoRAG-Synth(R-L)|Avg.|
> |-|-|-|-|-|-|-|-|-|-|
> |Ours|62.50|47.08|26.96|12.04|46.32|35.65|19.23|28.23|36.82|
> |-
> |DPR|61.50|44.14|24.11|8.86|46.32|35.55|19.23|28.23|35.18|
> |CLIP|62.00|47.08|26.63|11.71|44.70|32.52|18.53|21.30|34.70|
> |DPR+CLIP|61.50|44.14|23.16|8.29|44.28|32.43|18.53|21.28|32.95|
> |-
> |Qwen3-Embedding-4B|**63.00**|**47.86**|27.75|**14.86**|46.32|35.26|19.23|28.23|37.61|
> |VLM2Vec-V2|62.50|47.08|26.63|12.00|**47.13**|36.04|**20.05**|29.75|37.33|
> |Qwen3-Embedding-4B + VLM2Vec-V2|**63.00**|**47.86**|**27.85**|**14.86**|**47.13**|**36.14**|**20.05**|**29.77**|**38.37**|
>
> The results show that classical encoders such as DPR and CLIP provide noticeably weaker performance across tasks, while state-of-the-art encoders yield consistent improvements over the current configuration, with their combination achieving the strongest overall performance. This demonstrates that UniversalRAG can effectively benefit from high quality modality-specific encoders, which can be easily integrated into the framework.

---

> > ### Author Response · Authors · 2025-11-22
> > **Response to Reviewer Vohn (2/2)**
> >
> > > **W3. Cross-modal scenarios underrepresented**
> >
> > We acknowledge that cross-modal performance does not appear prominently in many of our current benchmarks. However, this limitation is not due to UniversalRAG itself, but rather to the lack of datasets designed for multimodal RAG, where most are predominantly designed so that each query can be answered using evidence from a single modality. Therefore, it is natural that datasets other than HybridQA and WebQA, which are explicitly designed to require cross-modal retrieval, do not show substantial benefits from cross-modal processing.
> >
> > Although there is no benchmark that truly requires retrieval across multiple independent modalities, a few multimodal datasets rely on interleaved sources (e.g., text paired with images) rather than separate modality-specific corpora. HybridQA and WebQA fall into this category, and we adapt both benchmarks to our setting following the procedure detailed in `Appendix A.1`. As shown in `Table 1`, UniversalRAG (Cross-GPT-4.1) achieves a 3.4% performance improvement over UniversalRAG (GPT-4.1) on these datasets, highlighting a clear gain when queries require multi-source aggregation. To further examine cross-modal capabilities, we also evaluated UniversalRAG on InfoSeek [1] and MRAG-Bench [2], where queries consist of text with images and retrieval across both modalities is beneficial. `Table A.3` reports results for both the uni-modal and cross-modal variants of UniversalRAG with GPT-4.1, along with representative baselines.
> >
> > **Table A.3. Evaluation of UniversalRAG and baselines on InfoSeek and MRAG-Bench**
> > |Retriever|InfoSeek (Acc)|MRAG-Bench (Acc)|
> > |-|-|-|
> > |Naive|21.68|43.68|
> > |ParagraphRAG|44.89|44.86|
> > |DocumentRAG|35.70|42.72|
> > |TableRAG|17.94|37.77|
> > |ImageRAG|32.39|51.22|
> > |ClipRAG|18.25|37.03|
> > |VideoRAG|18.86|36.73|
> > |InternVideo2|41.70|43.46|
> > |GME|42.25|45.16|
> > |All|32.88|45.01|
> > |-|
> > |UniversalRAG (GPT-4.1)|43.11|49.67|
> > |UniversalRAG (Cross-GPT-4.1)|**45.99**|**52.11**|
> >
> > The results show that UniversalRAG achieves strong performance on both InfoSeek and MRAG-Bench, outperforming a diverse set of baselines. More importantly, the cross-modal variant, UniversalRAG (Cross-GPT-4.1), consistently surpasses the uni-modal version, with improvements of 2.88% and 2.78% respectively. These gains demonstrate that when cross-modal knowledge is actually required, UniversalRAG can effectively leverage retrieval from multiple modalities.
> >
> > ---
> > [1] Can Pre-trained Vision and Language Models Answer Visual Information-Seeking Questions?, EMNLP 2023.
> >
> > [2] MRAG-Bench: Vision-Centric Evaluation for Retrieval-Augmented Multimodal Models, ICLR 2025.

---

### Note · Authors · 2025-12-08

I have read and agree with the venue's withdrawal policy on behalf of myself and my co-authors.